# Towards Addressing Label Skews in One-Shot Federated Learning

**Yiqun Diao, Qinbin Li & Bingsheng He**
National University of Singapore
`{yiqun,qinbin,hebs}@comp.nus.edu.sg`

## Abstract

Federated learning (FL) has been a popular research area, where multiple clients collaboratively train a model without sharing their local raw data. Among existing FL solutions, one-shot FL is a promising and challenging direction, where the clients conduct FL training with a single communication round. However, while label skew is a common real-world scenario where some clients may have few or no data of some classes, existing one-shot FL approaches that conduct voting on the local models are not able to produce effective global models. Due to the limited number of classes in each party, the local models misclassify the data from unseen classes into seen classes, which leads to very ineffective global models from voting. To address the label skew issue in one-shot FL, we propose a novel approach named FedOV which generates diverse outliers and introduces them as an additional *unknown* class in local training to improve the voting performance. Specifically, based on open-set recognition, we propose novel outlier generation approaches by corrupting the original features and further develop adversarial learning to enhance the outliers. Our extensive experiments show that FedOV can significantly improve the test accuracy compared to state-of-the-art approaches in various label skew settings. Code is available at https://github.com/Xtra-Computing/FedOV.

## 1 Introduction

Federated learning (FL) (McMahan et al., 2016; Kairouz et al., 2019; Yang et al., 2019; Li et al., 2019) allows multiple clients to collectively train a machine learning model while preserving individual data privacy. Most FL algorithms like FedAvg (McMahan et al., 2016) require many communication rounds to train an effective global model, which cause massive communication overhead, increasing privacy concerns, and fault tolerance requirements among rounds. One-shot FL (Guha et al., 2019; Li et al., 2021c), i.e., FL with only a single communication round, has been a promising and challenging direction to address the above issues.

On the other hand, label skews are common in real-world applications, where different clients have different label distributions (e.g., hospitals on different regions can face different diseases). As parties may have few or no data of some classes, this leads even more challenges in one-shot FL. In this paper, we study whether and how we can improve the effectiveness of one-shot FL algorithm for applications with label skews.

A simple and common one-shot FL strategy (Guha et al., 2019; Li et al., 2021c) is to conduct local training and collect the local models as an ensemble. The ensemble is either directly used as a final model for predictions (Guha et al., 2019) or distilled as a single model (Li et al., 2021c) with voting. However, those voting based approaches fail to produce high quality federated learning models.

Under the label skew setting, since each client has only a portion of classes, the local model predicts everything to its seen classes and the final voting results are poor. For example, in an extreme case where each client only has one label (e.g., face recognition), all clients predict the input as its own label and the voting result is meaningless. To address this issue, we propose *open-set voting* for one-shot FL that introduces an "unknown" class in the voting inspired by studies on open-set recognition (OSR) (Neal et al., 2018; Zhou et al., 2021). In local training, the clients train local open-set classifiers that are expected to predict its known classes correctly, while predicting "unknown" if it is unsure about the input data. Then, during inference, the server conducts voting on the received open-set

classifiers with the "unknown" option. In this way, open-set voting can filter the local models that do not have the knowledge of an input to improve the voting accuracy.

Although there are existing OSR studies in the centralized setting, it is challenging to apply them in the label-skewed federated setting to achieve good open-set classifiers during local training due to the limited number of local classes. For example, the state-of-the-art OSR algorithm PROSER (Zhou et al., 2021) considers linear interpolation between different seen classes as outliers. The outliers and the original data are used to train the model, where the outliers are considered as the extra class "unknown". When the number of classes is very small in a client, PROSER generates very limited types of outliers that are not sufficient for training. Moreover, the classifier has a loose boundary as the distance between the training data and the generated outliers may be large. To improve the quality of open-set classifiers, we propose a new open-set approach named FedOV with two novel techniques including data destruction and adversarial outlier enhancement to generate diverse and tight outliers. In data destruction, as opposed to data augmentation, we generate rich outliers by transforming the key features from the original image using boosted data operations such as random erasing and random resized crop. In adversarial outlier enhancement, we further optimize the outliers to be close to the training data in an adversarial way such that the local model cannot distinguish it.

Experiments show that our proposed FedOV (Federated learning by Open-set Voting) significantly improves the accuracy compared with existing one-shot FL approaches under various label skew cases. To reduce the model size of FedOV ensemble, we also combine knowledge distillation to FedOV like previous approaches (Lin et al., 2020; Li et al., 2021c). Distilled FedOV can also outperform state-of-the-art one-shot FL algorithms with model distillation.

Our main contributions are summarized as follow:

- To the best of our knowledge, we are the first to propose open-set voting in FL by introducing the "unknown" class, which significantly improves the accuracy compared to traditional close-set voting in FL.

- We propose two novel techniques, including data destruction and adversarial outlier enhancement, to generate diverse "unknown" outliers without requirement on the number of classes of the training data.

- We conduct extensive experiments to show the effectiveness of our open-set voting algorithm. Our algorithm consistently outperforms baselines with a significant improvement on accuracy on comprehensive label skew settings, including $\#C = 1$ (each client has only one class) where many FL algorithms fail.

## 2 BACKGROUND AND RELATED WORK

### 2.1 NON-IID DATA IN FL

Non-IID data is prevalent among real-world applications. For example, different areas have different types of diseases. Another example is that there are different species in different places. For classification tasks, suppose client $i$ has dataset $\{x_i, y_i\}$, where $x_i$ are features and $y_i$ are labels. In the label skew setting, $p(y_i)$ differs across clients.

Label skew is difficult because the local optima of different clients can be much far away (Li et al., 2021b). There have been many studies (Li et al., 2020; 2021d; Wang et al., 2020a; Luo et al., 2021; Mendieta et al., 2022) trying to alleviate this problem based on the model-averaging framework. For example, FedProx (Li et al., 2020) and MOON (Li et al., 2021d) adjusts the local training procedure to pull back local models from global model. FedNova (Wang et al., 2020a) normalizes local steps of each client during aggregation. A recent work (Huang et al., 2022) proposes few-shot model agnostic FL, which is able to train any models in a setting where each client has a very small sample size. It applies domain adaptation in the latent space with the help of a large public dataset. However, these algorithms need many rounds to converge, which may not be practical in real-world scenarios. For example, different companies may not be willing to communicate with each other frequently due to privacy and security concerns.

## 2.2 ONE-SHOT FL ALGORITHMS

One-shot FL (i.e., FL with only one communication) is a promising research direction. It has a very low communication cost. Moreover, it enables applications such as model market (Vartak, 2016), where the clients only need to sell the models to the market and users conduct learning on the models.

The original one-shot FL study (Guha et al., 2019) collects local models as an ensemble for the final prediction. It further proposes to use knowledge distillation on such ensemble with public data. FedKT (Li et al., 2021c) proposes consistent voting to improve the ensemble. A recent work (Zhang et al., 2021) proposes a data-free knowledge distillation scheme to perform one-shot FL. It uses the same ensemble distillation method as FedDF (Lin et al., 2020). Its main contribution is the fake data generation in the server side to replace the public dataset for distillation. Such a technique is orthogonal to our FedOV, and can be combined with FedOV. All the above studies do not investigate comprehensive label skew cases. For more related studies and a detailed comparison between FedOV and these studies, please refer to Appendix A.1.

## 2.3 OPEN-SET RECOGNITION (OSR)

OSR is an emerging field with many important applications (Salehi et al., 2021). In OSR, an additional class "unknown" is introduced besides the original classes. A good open-set classifier is expected to predict its known classes correctly, while predicting "unknown" if it is unsure about the input data. A popular approach in OSR is to generate outliers and label them as "unknown" in training.

One direction is to use GANs to generate outliers. For example, Neal et al. (2018) apply GANs to generate outliers from latent space that (1) is close to real samples, and (2) with high probability of outlier (low probability of any known class). However, the training of GANs is very expensive. Moreover, it is challenging for GANs to generate clear high-dimension images.

PROSER (Zhou et al., 2021) is a state-of-the-art open-set recognition approach. It generate outliers by linear interpolation of embedding space among different classes. Moreover, it introduces an additional loss to increase the possibility of predicting a sample as "unknown" when discarding its true class. Suppose the training set is $D = \{(x_i, y_i)\}_{i=1}^n$, where $x_i$ is a training sample and $y_i \in \{0, 1, ..., c-1\}$ is its label. The neural network from input space to embedding space is denoted $\phi_{pre}(\cdot)$, and $\phi_{post}(\cdot)$ is the neural network from embedding space to output space. The whole model is $f(\cdot) = \phi_{post}(\phi_{pre}(\cdot))$ The PROSER loss is shown in Equation 1, where $l$ is the cross entropy loss, $c$ denotes class "unknown", $\tilde{x}_{pre}$ is the outliers generated from the training data $(\tilde{x}_{pre} = \lambda\phi_{pre}(x_i) + (1-\lambda)\phi_{pre}(x_j))$, $\lambda \in [0, 1]$ is sampled from Beta distribution, $\beta$ and $\gamma$ are two hyper-parameters.

$$L_{PROSER} = \sum_{(x,y) \in D} l(f(x), y) + \beta l(f(x)\backslash y, c) + \gamma \sum_{(x_i, x_j) \in D} l(\phi_{post}(\tilde{x}_{pre}), c) \tag{1}$$

Since PROSER achieves state-of-the-art results and easy to implement, we consider it as our base method for training an open-set classifier.

## 3 FEDOV: ONE-SHOT FEDERATED OPEN-SET VOTING FRAMEWORK

### 3.1 PROBLEM STATEMENT

Suppose there are $N$ clients $P_1, ..., P_N$ with local datasets $D^1, ..., D^N$. Our goal is to train a good machine learning model over $D \triangleq \bigcup_{i \in [N]} D^i$ with the help of a server, while the raw data are not exchanged. Moreover, each client is allowed to communicate with the server only once. In this paper, we focus on image classification task due to its popularity.

### 3.2 MOTIVATION

**Observation 1** Voting is a popular method in existing one-shot FL approaches (Guha et al., 2019; Li et al., 2021c). However, these approaches suffer under extreme label skews. For example, when we divide MNIST dataset into 10 clients where each client has only one class, both close-set voting

(Guha et al., 2019) and FedKT (Li et al., 2021c) only have lower than 20% test accuracy. When each client has two classes, the test accuracy of both methods are lower than 50%. The problem is that the predictions of close-set classification models are biased towards their seen classes as shown in Figure 1a. When a test sample of class one comes, in traditional close-set voting, the first and third client make wrong predictions. Therefore, the voting result cannot predict correctly.

**Implication 1**    In the label skew setting of FL, close-set classifiers are weak for one-shot FL and predict every input among its known classes. For voting, it would be better if models can be modest and admit unknown for its unseen classes as shown in Figure 1b. This motivates us to apply OSR in FL to introduce an unknown class to improve the voting.

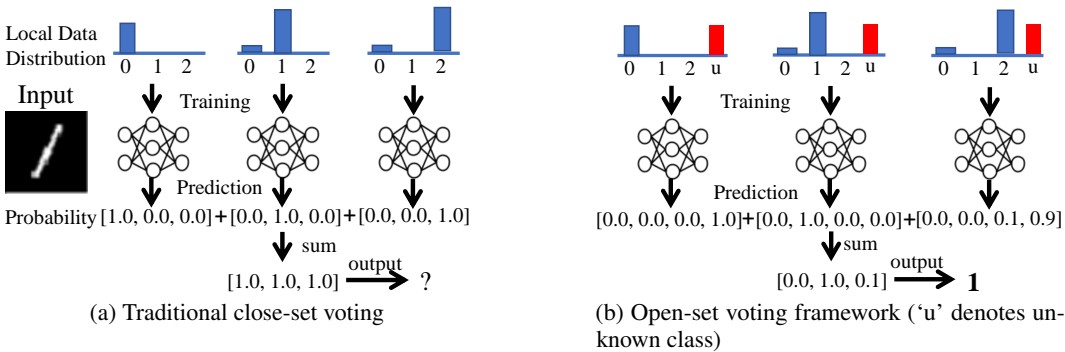

(a) Traditional close-set voting

(b) Open-set voting framework ('u' denotes unknown class)

Figure 1: Comparison between traditional close-set voting and our open-set voting framework. We allocate the data of classes 0, 1, and 2 into three clients by Dirichlet distribution $Dir_3(0.02)$.

**Observation 2**    Directly applying PROSER (Zhou et al., 2021) in the local training of FL cannot achieve good local open-set classifiers. We visualize the representation learned by the local model of a client using PROSER in the local training in Figure 2a. The model is trained on a client which only has class 0 and 6 samples of MNIST. The generated outliers are quite limited and far from the training data when simply applying PROSER. The representations from the data of the seen and unseen classes are mixed and cannot be distinguished.

**Implication 2**    To better suit OSR algorithms for label skews in FL, we need new techniques to generate outliers which should 1) be diverse and 2) be close to the seen classes. We will introduce them in Section 3.4 and Section 3.5 and explain Figure 2b and Figure 2c in Section 3.6.

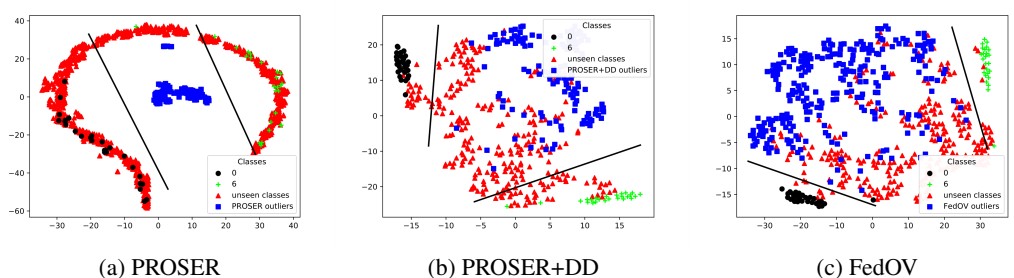

(a) PROSER

(b) PROSER+DD

(c) FedOV

Figure 2: T-SNE visualisation of the features extracted by local models trained with different methods. During training, the client only has data from classes 0 and 6. In each sub-figure, we plot the representations of the data from the seen classes 0 and 6, unseen classes, and generated outliers during training. The black lines are possible classification boundaries.

### 3.3 THE OVERALL ALGORITHM

Based on the above observations and implications, we develop a new approach named FedOV to address label skews in federated learning. FedOV addresses the challenges in directly applying OSR to FL in the following two aspects. First, in order to generate diverse outliers, we propose data destruction (DD) to directly generate outliers from true samples. Second, in order to generate outliers that are even closer to true samples, we propose adversarial outlier enhancement (AOE) to learn a tighter boundary to surround the inliers.

The overall framework of open-set voting is described as follows. For the training stage, each client trains an open-set classifier locally and submits it to the server. For the prediction stage, the server sums up the prediction probability of all submitted models on the input sample while discarding their "unknown" channel. The class with maximum prediction probability is outputted as the prediction label. An example of open-set voting is shown in Figure 1b. With the help of the unknown class, local models can admit its uncertainty when encountering unseen classes. The first and third model with little knowledge of class 1 assign very high probability to "unknown", and the second model outputs class 1 with 100% certainty due to its expertise in class 1. In this way, the input image can be correctly classified.

The whole procedure is shown in Algorithm 1. Suppose there are $c$ classes and classes 0 to $(c-1)$ are the classes from the original training data. We use class $c$ to denote the unknown class. In each client, it first initializes the local model $f_i$ (Line 2). Then, in each round, for each batch of training data, it generate outliers by data destruction and adversarial outlier enhancement (Lines 5-6, see Section 3.4 and 3.5 for more details). Next, considering the outliers as the unknown class, cross-entropy loss is computed on the outliers. By summing the PROSER loss (computed according to Equation 1) and the cross-entropy loss as the total loss, the local model is updated using the Adam optimizer (Lines 7-8). The local models are sent to server after reaching the number of training rounds (Lines 9).

In the server, it aggregates all the local models as an ensemble as the final model (Line 11). When a new sample comes for prediction, it sums the prediction probability of each model (Lines 13-15). Then, the known class with the highest probability score is considered as the output label (Line 16).

Since FedOV only requires a single communication round, its communication cost is $O(NM)$, where $M$ is the size of local model. The communication cost is low compared with iterative federated learning algorithms, which need many rounds to communicate the models.

---

**Algorithm 1:** The FedOV algorithm. $L_{ce}$ is the cross entropy loss and $\sigma$ is the softmax function.

**Input:** number of clients $N$, number of classes $c$, training rounds $T$

1  **Each client executes**:
2  Initialize local model $f_i$
3  **for** $t = 1, ..., T$ **do**
4     **for** each batch of local data $(x, y)$ **do**
5        $x' = DataDestruction(x)$
6        $x'' = FGSM(f_i, x', c)$
7        $L = L_{PROSER} + L_{ce}(f_i(\{x', x''\}), c)$
8        Update $f_i$ with loss $L$
9  Upload $f_i$ to the server.

10  **Server executes**:
11  Collects $f_1, ... f_N$ as an ensemble.
12  **Prediction**$(x)$:
13  $scores = \mathbf{0}$
14  **for** $i = 1, ..., N$ **do**
15     $scores = scores + \sigma(f_i(x))$
16  $y_p = \arg\max_{j \in \{0,1,2,...,c-1\}} scores_j$
17  **return** $y_p$

---

### 3.4 DATA DESTRUCTION

From our observation in Section 3.2, generating diverse outliers from limited training data is challenging. While PROSER generates outliers by mixing data from different classes, can we generate outliers from each individual sample? Inspired by data augmentation (Shorten & Khoshgoftaar, 2019) which has been a very popular approach to enhance the features before training, we propose the novel Data Destruction (DD) methods to use data operations to transform the data to generate outliers. As opposed to applying to enhance the features, our DD applies intense data operations to corrupt the original key features, which is effective and efficient. Specifically, DD has two components: candidate data destruction operations and boosting outliers with a set of such operations.

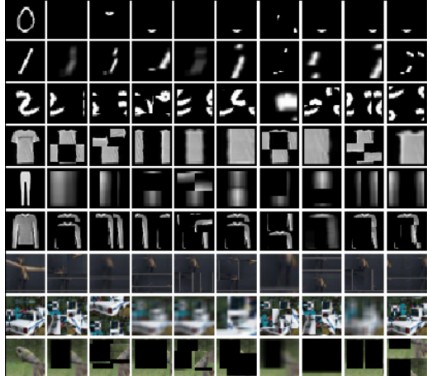
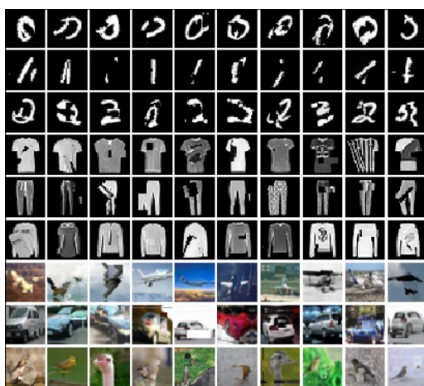

(a) Outliers generated by DD.                    (b) Outliers after AOE.

Figure 3: Generated outlier examples on MNIST, Fashion-MNIST and CIFAR-10 dataset. In Figure 3a, the first column contains the real samples and the other columns are generated outliers of that sample. In Figure 3b, all figures are generated outliers by adversarial outlier enhancement.

**Candidate Data Destruction Operations**    With the goal to corrupt the original features, we try a comprehensive list of data operations and summarize the following useful candidate operations for data destruction: (1) RandomCopyPaste: randomly select a rectangle region and copy it to another randomly selected region in an image; (2) RandomSwap: swap two randomly selected rectangle regions; (3) RandomRotation: randomly rotate a square region of an image; (4) RandomErasing: randomly erasing a large rectangle region in an image; (5) GaussianBlur: blur an image by a Gaussian function with a large variance; (6) RandomResizedCrop: randomly crop a small portion of an image and resize it to the original size. Here (1)-(3) are our proposed operations and (4)-(6) are existing data augmentation operations where we use them with abnormal hyper-parameters. Examples of the generated outliers are shown in Figure 3a. More details about these are elaborated in Appendix B.1.

**Boosting Outliers with Data Destruction Set**    To boost the diversity of outliers, we introduce randomness during the outlier generation. We do not use a fixed operation on each image to generate outliers. In each time, considering the above candidate operations as a set, we randomly sample one operation to generate an outlier each time. Then, in each batch of data during training, there exists diverse types of outliers generated by different operations. We show that such boosting method can significantly improve the accuracy in Appendix B.3.

In summary, the intuition behind DD is to corrupt part of the key features such that 1) the generated data are not similar to the training data and 2) the generated data is diverse so that the model does not simply consider certain patterns as outliers.

## 3.5 ADVERSARIAL OUTLIER ENHANCEMENT

Adversarial training (Goodfellow et al., 2015; Kurakin et al., 2016) has been a popular approach to protect machine learning models from malicious attacks. For example, Goodfellow et al. (2015) utilizes fast gradient sign method (FGSM) to generate adversarial samples such that the model outputs a wrong answer with a high confidence. Then, the adversarial samples are used as a part of the training data to regularize the training.

Inspired by adversarial training, instead of using FGSM to generate adversarial samples for robust training, we apply it to optimize the generated outliers. Specifically, suppose the client is training the model $f$ with the generated outliers $x'$ by our data destruction method. We utilize FGSM to generate $x''$ such that the model wrongly outputs $x''$ as a seen sample with a high confidence. Then, the enhanced outliers $x''$ are used together with the generated outliers $x'$ as the unknown class to update the model. We call this method Adversarial Outlier Enhancement (AOE). Examples of the enhanced outliers are shown in Figure 3b. Compared Figure 3b with Figure 3a, the outliers are more normal and look like different classes from the training data after AOE (e.g., in the third row of Figure 3b, the third and eight outliers look like digit "3" although they are generated from digit "2".).

### 3.6 DISCUSSION

**T-SNE Visualization**   As shown in Figure 2b and Figure 2c, DD can generate diverse outliers to help distinguish data from seen and unseen classes and AOE can further decrease the margin between the outliers and training data (i.e., class 0 and 6) to learn a better classifier.

**Extension of FedOV with Knowledge Distillation**   One shortage of FedOV is that the final model is an ensemble of local models, therefore its prediction and storage costs may be large especially when the number of clients is large (e.g., cross-device setting). Some existing approaches (Lin et al., 2020; Li et al., 2021c) utilize knowledge distillation to distill the knowledge from multiple local models to a global model with the help of a public or synthetic dataset. Our approach is compatible with the above methods. With knowledge distillation, we can transform the ensemble of local models into a single global model. Then, it can significantly reduce the storage and prediction costs of the final model. Moreover, considering the final model as the initialized model for iterative federated learning algorithms like FedAvg (McMahan et al., 2016), we can conduct multi-round federated learning to further improve the model. As shown in Section 4.4, FedOV can be effectively combined with the existing approaches to increase their accuracy and communication efficiency.

## 4   EXPERIMENTS

### 4.1   EXPERIMENTAL DETAILS

**Datasets**   We conduct experiments on MNIST, Fashion-MNIST, CIFAR-10 and SVHN datasets. We use the data partitioning methods in Li et al. (2021b) to simulate different label skews. Specifically, we try two different kinds of partition: 1) $\#C = k$: each client only has data from $k$ classes. We first assign $k$ random class IDs for each client. Next, we randomly and equally divide samples of each class to their assigned clients; 2) $p_k \sim Dir(\beta)$: for each class, we sample from Dirichlet distribution $p_k \sim Dir_N(\beta)$ and distribute $p_{k,j}$ portion of class $k$ samples to client $j$.

**Baselines**   We include one-shot FL algorithms as baselines including close-set voting (Guha et al., 2019) and FedKT (Li et al., 2021c). We also compare FedOV with the iterative FL algorithms including FedAvg (McMahan et al., 2016), FedProx (Li et al., 2020), FedNova (Wang et al., 2020a), and FedDF (Lin et al., 2020). We run them in a single round for a fair comparison. Note that FedKT and FedDF require a public dataset (or synthetic dataset) for distillation. In each task, we use a half of the test dataset as the public dataset for distillation for FedKT and FedDF and the remaining for testing. Since the source code of FedSyn (Zhang et al., 2021) is not publicly available and we have included FedDF (which has the same distillation approach as FedSyn) in our experiments, we do not compare it with FedOV.

**Default setups**   By default, we follow FedAvg (McMahan et al., 2016) and other existing studies (Li et al., 2021c;b; Wang et al., 2020b) to use a simple CNN with 5 layers in our experiments. There are 10 clients. For local training, we run 200 local epochs for each client. We set batch size to 64 and learning rate to 0.001. For results with error bars, we run three experiments with different random seeds.

Due to the page limit, we only present some representative results in the main paper. For more experimental details and results, please refer to Appendix B.

### 4.2   AN OVERALL COMPARISON

We compare the accuracy between FedOV and the other baselines as shown in Table 1. Our algorithm can significantly outperform baseline algorithms with only one communication. In many settings, FedOV achieves more than 10% accuracy than close-set voting. In the extreme cases such as $\#C = 1$, FedOV can outperform close-set voting by at lease 30%. The iterative FL algorithms cannot achieve satisfactory accuracy when running for a single round.

Table 1: Comparison with close-set voting and various FL algorithms in one round.

| Dataset | Partition | FedOV | Close-set voting | FedAvg | FedProx | FedNova | SCAFFOLD | FedDF | FedKT |
|---|---|---|---|---|---|---|---|---|---|
| CIFAR-10 | #C = 1 | **40.0%±1.7%** | 10.2%±0.2% | 10.5%±1.0% | 10.6%±1.3% | 10.5%±1.0% | 10.5%±1.0% | 10.2%±0.5% | 9.8%±0.2% |
| | #C = 2 | **42.0%±2.4%** | 37.2%±2.5% | 11.1%±1.9% | 10.9%±1.6% | 10.5%±0.7% | 11.1%±1.8% | 18.8%±1.1% | 25.7%±2.9% |
| | #C = 3 | **55.6%±6.3%** | 43.2%±2.7% | 15.7%±5.1% | 15.9%±5.3% | 14.5%±3.9% | 16.1%±5.0% | 27.5%±4.0% | 31.8%±2.5% |
| | $p_k \sim Dir(0.5)$ | **65.7%±0.7%** | 65.0%±0.1% | 18.4%±7.2% | 18.7%±5.3% | 19.8%±7.2% | 18.6%±5.1% | 35.3%±0.9% | 42.1%±2.5% |
| | $p_k \sim Dir(0.1)$ | **61.7%±1.1%** | 55.9%±1.3% | 10.4%±0.4% | 11.1%±0.9% | 13.1%±3.3% | 13.0%±4.6% | 26.3%±3.0% | 35.0%±1.7% |
| SVHN | #C = 1 | **64.5%±1.9%** | 7.3%±0.1% | 19.1%±0.9% | 18.3%±2.2% | 10.4%±2.5% | 18.9%±1.2% | 7.0%±0.7% | 6.6%±0.1% |
| | #C = 2 | **74.0%±1.1%** | 35.7%±7.6% | 19.0%±4.1% | 17.0%±6.7% | 14.4%±4.9% | 16.2%±9.0% | 54.3%±2.8% | 30.9%±4.3% |
| | #C = 3 | **81.0%±1.2%** | 60.1%±9.5% | 24.9%±6.2% | 24.1%±4.9% | 17.3%±1.2% | 19.3%±4.2% | 62.1%±4.3% | 59.7%±2.5% |
| | $p_k \sim Dir(0.5)$ | **85.7%±0.3%** | 85.1%±0.4% | 32.4%±9.5% | 33.5%±10.4% | 33.6%±9.7% | 29.3%±4.0% | 80.7%±1.6% | 75.7%±1.6% |
| | $p_k \sim Dir(0.1)$ | **78.7%±0.6%** | 64.9%±0.4% | 20.1%±0.4% | 21.4%±2.1% | 22.3%±2.8% | 20.7%±1.1% | 68.0%±0.8% | 54.6%±1.9% |
| FMNIST | #C = 1 | **73.3%±1.6%** | 10.1%±0.5% | 13.1%±5.4% | 13.2%±5.5% | 13.1%±5.4% | 13.1%±5.4% | 12.1%±1.6% | 9.8%±0.1% |
| | #C = 2 | **61.7%±11.0%** | 36.3%±5.5% | 23.1%±5.4% | 23.2%±3.9% | 17.7%±2.7% | 22.1%±6.6% | 37.0%±11.3% | 28.8%±6.0% |
| | #C = 3 | **73.8%±1.7%** | 57.0%±7.0% | 26.1%±2.0% | 26.8%±0.3% | 24.9%±4.3% | 26.0%±1.4% | 46.7%±11.2% | 51.2%±7.4% |
| | $p_k \sim Dir(0.5)$ | **88.9%±0.3%** | 88.5%±0.2% | 55.1%±12.6% | 54.2%±13.4% | 54.1%±7.9% | 52.6%±10.0% | 83.0%±2.2% | 80.9%±2.4% |
| | $p_k \sim Dir(0.1)$ | **76.2%±1.2%** | 73.6%±0.4% | 22.6%±14.5% | 24.2%±17.6% | 24.8%±15.2% | 19.0%±7.7% | 66.4%±6.8% | 54.2%±8.5% |
| MNIST | #C = 1 | **79.3%±1.8%** | 15.5%±2.8% | 10.1%±1.2% | 10.1%±1.2% | 10.1%±1.2% | 10.1%±1.2% | 11.4%±0.3% | 9.9%±0.4% |
| | #C = 2 | **64.2%±1.6%** | 44.3%±6.4% | 16.7%±6.7% | 12.7%±3.9% | 20.9%±12.0% | 12.0%±2.8% | 53.1%±4.0% | 33.8%±8.1% |
| | #C = 3 | **83.7%±5.3%** | 59.6%±7.0% | 29.8%±19.0% | 29.9%±19.0% | 24.2%±13.5% | 26.5%±18.3% | 71.4%±6.0% | 55.0%±12.2% |
| | $p_k \sim Dir(0.5)$ | **98.6%±0.0%** | 98.3%±0.1% | 67.5%±2.8% | 71.6%±9.3% | 74.3%±6.9% | 67.7%±2.2% | 97.9%±0.3% | 94.9%±0.5% |
| | $p_k \sim Dir(0.1)$ | **96.2%±0.4%** | 93.3%±0.4% | 40.2%±5.6% | 39.7%±6.5% | 40.1%±4.7% | 35.3%±7.3% | 82.8%±7.4% | 68.0%±13.1% |

Table 2: Experimental results of different FL voting strategies with simple CNN model.

| Dataset | Partition | Close-set | Open-set (PROSER) | Open-set (PROSER + DD) | FedOV |
|---|---|---|---|---|---|
| CIFAR-10 | #C = 1 | 10.2%±0.2% | 10.6%±0.2% | 33.5%±2.3% | **40.0%±1.7%** |
| | #C = 2 | 37.2%±2.5% | 34.8%±4.5% | 41.3%±7.7% | **42.0%±2.4%** |
| | #C = 3 | 43.2%±2.7% | 50.2%±4.7% | 54.3%±2.1% | **55.6%±6.3%** |
| | $p_k \sim Dir(0.5)$ | 65.0%±0.1% | 66.6%±0.1% | **67.6%±0.3%** | 65.7%±0.7% |
| | $p_k \sim Dir(0.1)$ | 55.9%±1.3% | 58.0%±0.9% | 61.3%±1.0% | **61.7%±1.1%** |
| SVHN | #C = 1 | 7.3%±0.1% | 6.7%±0.1% | 47.3%±1.3% | **64.5%±1.9%** |
| | #C = 2 | 35.7%±7.6% | 42.6%±10.9% | 60.9%±1.7% | **74.0%±1.1%** |
| | #C = 3 | 60.1%±9.5% | 64.9%±8.4% | 72.7%±1.3% | **81.0%±1.2%** |
| | $p_k \sim Dir(0.5)$ | 85.1%±0.4% | 85.2%±0.3% | 84.9%±0.3% | **85.7%±0.3%** |
| | $p_k \sim Dir(0.1)$ | 64.9%±0.4% | 65.9%±0.8% | 74.6%±0.8% | **78.7%±0.6%** |
| FMNIST | #C = 1 | 10.1%±0.5% | 15.1%±1.1% | 71.0%±2.1% | **73.3%±1.6%** |
| | #C = 2 | 36.3%±5.5% | 33.0%±1.7% | **64.1%±6.7%** | 61.7%±11.0% |
| | #C = 3 | 57.0%±7.0% | 51.9%±0.9% | 66.1%±2.4% | **73.8%±1.7%** |
| | $p_k \sim Dir(0.5)$ | 88.5%±0.2% | 88.7%±0.2% | **89.1%±0.1%** | 88.9%±0.3% |
| | $p_k \sim Dir(0.1)$ | 73.6%±0.4% | 73.2%±0.9% | 76.0%±0.8% | **76.2%±1.2%** |
| MNIST | #C = 1 | 15.5%±2.8% | 16.5%±0.2% | 76.5%±8.3% | **79.3%±1.8%** |
| | #C = 2 | 44.3%±6.4% | 48.7%±4.2% | 61.5%±9.3% | **64.2%±1.6%** |
| | #C = 3 | 59.6%±7.0% | 55.9%±1.2% | 73.3%±2.6% | **83.7%±5.3%** |
| | $p_k \sim Dir(0.5)$ | 98.3%±0.1% | 98.3%±0.1% | 98.5%±0.1% | **98.6%±0.0%** |
| | $p_k \sim Dir(0.1)$ | 93.3%±0.4% | 93.8%±0.5% | 95.8%±0.2% | **96.2%±0.4%** |

## 4.3 ABLATION STUDY

We show the effect of each component of FedOV including open-set voting (PROSER), data destruction (DD), and adversarial outlier enhancement (AOE). Specifically, we add one component each time and the results are shown in Table 2. From the table, we can observe that FedOV with all the three components can achieve the highest accuracy in most settings. Simply applying PROSER in FL may not increase the accuracy compared with close-set voting (e.g., CIFAR-10 with $\#C = 2$). Our proposed outlier generation techniques can effectively boost the accuracy of open-set voting. Moreover, the adversarial outlier enhancement can significantly increase the accuracy in some settings (e.g., SVHN with $\#C = 1$). We compare with open-set voting (PROSER + AOE), i.e., using AOE loss without DD loss in Appendix B.2.

## 4.4 COMBINING WITH KNOWLEDGE DISTILLATION

Assuming that there exist unlabeled public data on the server, FedOV can also be combined with knowledge distillation like FedDF (Lin et al., 2020) and FedKT (Li et al., 2021c). We call it Distilled FedOV. We compare Distilled FedOV with the other baselines. According to FedKT experimental settings, we train 100 epochs for each client. For distillation, we run 100 epochs for the final student model. According to our default settings, we use the simple CNN model and use SVHN without

Table 3: Comparing distilled FedOV with the other baselines. The partition is $p_k \sim Dir(0.5)$.

| Dataset | Distilled FedOV | FedKT | FedDF | SOLO | FedAvg | FedProx | FedNova | SCAFFOLD |
|---|---|---|---|---|---|---|---|---|
| MNIST | **97.7%±0.3%** | 95.4%±0.8% | 97.1%±0.3% | 78.4%±2.7% | 67.3%±4.0% | 67.2%±3.9% | 69.6%±2.5% | 67.9%±4.2% |
| SVHN | **81.8%±1.4%** | 75.7%±1.0% | 80.6%±3.2% | 46.1%±4.2% | 28.1%±5.6% | 27.7%±4.9% | 27.2%±6.2% | 29.2%±5.0% |
| FMNIST | **85.0%±0.1%** | 82.5%±0.4% | 80.6%±3.3% | 62.3%±1.3% | 49.5%±12.9% | 48.7%±13.4% | 49.4%±11.3% | 48.9%±13.5% |
| CIFAR-10 | **51.6%±1.4%** | 41.5%±2.3% | 34.1%±1.9% | 28.4%±1.5% | 15.7%±4.3% | 15.6%±4.0% | 16.9%±4.1% | 16.5%±5.1% |

extended dataset. Note that the settings are different from FedKT paper (Li et al., 2021c), therefore our reported accuracy differs from FedKT paper. Further details are elaborated in Appendix B.1.

We run distilled FedOV for three times and results are shown in Table 3. We can observe that Distilled FedOV can achieve a higher accuracy than FedKT and the other iterative FL baselines with the same size of final model, which further verifies the effectiveness of our open-set voting framework.

**Extension to Multiple Rounds**   After knowledge distillation, we can further train the distilled model by FL averaging algorithms (e.g., FedAvg, FedProx, etc). We conduct experiments on MNIST with 10 clients and data partitioning $p_k \sim Dir(0.5)$. The results are shown in Figure 4. For FedOV_FedProx and FedKT_FedProx, we run Distilled FedOV and FedKT for the first round and using the global model as the initialized model for the later rounds using FedProx. From the figure, with the help of FedOV, the accuracy after first round is high. Then, FedOV_FedProx can converge much faster than the other algorithms.

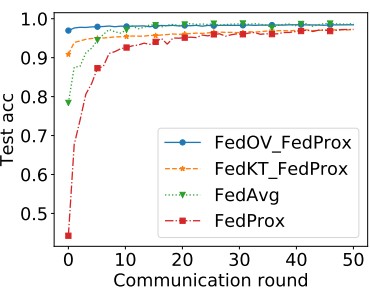

Figure 4: Extension to multiple rounds on MNIST.

## 4.5 SCALABILITY

We test the scalability of FedOV by varying the number of clients. Due to page limit, we only show results on CIFAR-10 in Table 4. For the results on the other datasets, please refer to Appendix B.8. From the table, we can observe that FedOV still achieves the best accuracy when increasing the number of clients. Moreover, with the help of knowledge distillation, Distilled FedOV can outperform distilled close-set and other iterative FL algorithms with the same storage and inference cost.

Table 4: Experimental results of different number of clients on CIFAR-10 with simple CNN model.

| Client Number | Partition | FedOV | Distilled FedOV | Close-set | Distilled close-set | FedAvg | FedProx | FedNova | SCAFFOLD | FedDF | FedKT |
|---|---|---|---|---|---|---|---|---|---|---|---|
| 20 | $\#C = 1$ | **41.9%** | 30.4% | 7.9% | 7.9% | 10.0% | 10.0% | 10.0% | 10.0% | 9.9% | 10.0% |
| | $\#C = 2$ | **45.6%** | 34.9% | 33.0% | 26.6% | 10.7% | 15.0% | 10.0% | 15.4% | 26.4% | 31.5% |
| | $\#C = 3$ | **57.2%** | 40.2% | 54.1% | 37.6% | 10.2% | 11.5% | 10.1% | 16.0% | 13.8% | 36.8% |
| | $p_k \sim Dir(0.5)$ | **62.5%** | 46.4% | 59.8% | 43.5% | 14.3% | 23.4% | 14.1% | 16.9% | 32.7% | 39.7% |
| | $p_k \sim Dir(0.1)$ | **52.4%** | 40.1% | 48.2% | 35.1% | 10.1% | 13.1% | 13.9% | 14.2% | 27.5% | 26.2% |
| 40 | $\#C = 1$ | **45.3%** | 34.3% | 10.1% | 9.6% | 10.0% | 10.0% | 10.0% | 10.0% | 8.5% | 10.0% |
| | $\#C = 2$ | **56.0%** | 40.4% | 42.1% | 32.3% | 10.0% | 10.5% | 10.0% | 11.1% | 27.2% | 26.3% |
| | $\#C = 3$ | **60.8%** | 45.2% | 52.1% | 41.9% | 10.0% | 10.3% | 10.0% | 12.7% | 29.8% | 35.8% |
| | $p_k \sim Dir(0.5)$ | **59.2%** | 46.7% | 58.2% | 46.7% | 11.6% | 18.7% | 14.2% | 18.4% | 32.3% | 37.2% |
| | $p_k \sim Dir(0.1)$ | **55.0%** | 41.7% | 48.3% | 40.4% | 10.3% | 16.5% | 10.4% | 14.9% | 26.4% | 26.2% |
| 80 | $\#C = 1$ | **43.8%** | 33.2% | 10.0% | 10.2% | 9.7% | 9.5% | 9.8% | 8.7% | 9.8% | 10.1% |
| | $\#C = 2$ | **56.6%** | 43.1% | 44.2% | 34.4% | 10.6% | 10.1% | 10.0% | 10.3% | 22.7% | 27.5% |
| | $\#C = 3$ | **54.1%** | 44.8% | 53.4% | 44.4% | 14.7% | 16.6% | 17.7% | 12.7% | 33.4% | 33.2% |
| | $p_k \sim Dir(0.5)$ | **53.4%** | 42.6% | 52.5% | 44.5% | 21.1% | 23.9% | 23.3% | 11.4% | 31.3% | 30.4% |
| | $p_k \sim Dir(0.1)$ | **48.5%** | 39.0% | 44.2% | 37.0% | 10.9% | 23.0% | 22.4% | 11.3% | 24.2% | 25.7% |

## 5 CONCLUSION

In this work, we design a novel one-shot FL algorithm FedOV to address label skews in one-shot federated learning. We propose open-set voting by introducing the "unknown" class in voting. We observe that directly applying state-of-the-art open-set recognition algorithm PROSER to one-shot learning has the problem of limited outliers due to limited number of classes. To address those issues, we develop two techniques, data destruction and adversarial outlier enhancement, to improve the performance of open-set voting. Our extensive experiments show that FedOV can achieve significant accuracy improvement compared with the other baselines under diverse label skew settings.

## ACKNOWLEDGEMENTS

This research is supported by the National Research Foundation, Singapore under its AI Singapore Programme (AISG Award No: AISG2-RP-2020-018). Any opinions, findings and conclusions or recommendations expressed in this material are those of the authors and do not reflect the views of National Research Foundation, Singapore.

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

# A Discussions

## A.1 More related works on one-shot FL

From the perspective of data sharing, Zhou et al. (2020) proposes to perform dataset distillation and upload the distilled data to server for centralized training. Kasturi et al. (2020) proposes to fit the features in each client by some distribution. Then the server generates fake data with these distributions. Both approaches raise additional privacy concerns due to the uploaded fake data or feature distributions. Besides, we cannot find source code for both methods, so we do not compare FedOV with them.

XorMixFL (Shin et al., 2020) proposes to use exclusive OR operation (XOR) to encode and decode samples for data sharing. However, it assumes that all clients and the server have labelled samples of a global class, which is impractical in real-world applications. In our experiments, we adopt the setting like many existing studies (Li et al., 2020; Karimireddy et al., 2020; Wang et al., 2020a; Hsu et al., 2019; Li et al., 2021b), which cannot guarantee the above assumption. Therefore, we do not compare FedOV with XorMixFL in our experiments.

We compare these one-shot FL algorithms in Table 5. None of the previous one-shot FL algorithms conduct experiments on both distribution-based and quantity-based label skews.

Table 5: Comparison among existing one-shot FL algorithms.

| Perspective | Algorithm | Distribution-based label skews | Quantity-based label skews | Only model uploaded | Free of public data |
|---|---|---|---|---|---|
| Model ensemble | Close-set voting (Guha et al., 2019) | ✗ | ✗ | ✓ | ✓ |
| | Distilled Close-set (Guha et al., 2019) | ✗ | ✗ | ✓ | ✗ |
| | FedKT (Li et al., 2021c) | ✓ | ✗ | ✓ | ✗ |
| | FedSyn (Zhang et al., 2021) | ✓ | ✗ | ✓ | ✓ |
| | FedOV (ours) | ✓ | ✓ | ✓ | ✓ |
| | Distilled FedOV (ours) | ✓ | ✓ | ✓ | ✗ |
| Sharing fake data | DOSFL (Zhou et al., 2020) | ✗ | ✓ | ✗ | ✓ |
| | Fusion learning (Kasturi et al., 2020) | ✗ | ✗ | ✗ | ✓ |
| | XorMixFL (Shin et al., 2020) | ✗ | ✓ | ✗ | ✗ |

## A.2 Implications for the Difficulty of Generating Outliers by GANs

While we propose data destruction to generate outliers, another method is to generate outliers by GANs. We apply the methods in Neal et al. (2018) to generate counterfactual images in MNIST dataset. However, it is very difficult to tune the hyper-parameters to generate outliers. We encounter two problems: (1) generated images are so good that the classifier goes on strike because it cannot tell the difference between original images and counterfactual images; (2) generated images are so bad that the classifier can easily tell the difference by some simple features. The loss quickly converges to zero. For example, since the generated counterfactual images are all blurry, the classifier may classify clear images as inlier and blurry images as outlier. When it encounters clear images of other digits during testing, it can classify these samples into its known class.

By trials and errors, we summarize two criteria of good outliers for future researchers' reference. Good outlier needs to (1) be clearly different from inliers; and (2) not carry certain types of simple feature that can be easily told difference.

## A.3 Challenges of Applying OSR to FL and Possible Directions

One possible future direction of FedOV is to set a threshold to normalize each client's outlier probability. Take a simple example, if a client only sees dogs, another client only sees cats. The first model predicts 0.001% outlier for dogs and 0.01% outlier for cats. The second model predicts 5% outlier for cats and 50% outlier for dogs. For ROC-AUC metric, both models get 100% ROC-AUC for outlier detection among dogs and cats. However for FL open-set voting framework, the voting result is always dog. This is due to the unreasonable outlier probability scale of first model.

Another promising direction is to come up with more systematic and diverse data destruction methods. Although our current approach can outperform cutpaste, the outlier types are still limited. For future research, one can adapt more computer vision algorithms to further improve diversity.

Finally, our current data destruction framework only works for vision tasks. For tabular dataset or language dataset, it deserves further research how to augment diverse outliers.

### A.4 A BETTER BENCHMARK FOR OSR ALGORITHMS

From the perspective of open-set recognition, we find that FL voting can serve as a more realistic, comprehensive and challenging benchmark to test open-set recognition algorithms. Previous open-set recognition experiments mainly divide the classes into totally known and totally unknown, i.e. the training set has either full data of a class or no data of a class. However, in reality, grey area exists where a model sees only a few training samples of some class. Moreover, previous open-set recognition or outlier detection experiments use ROC-AUC as the metric, which avoids transforming the outlier score to some probability. Instead, ROC-AUC only cares the relative score between known and unknown class. However, it is essential to output a reasonable unknown probability in real-world application. Our FL open-set voting benchmark includes these challenges. Its data partitioning strategies are based on Li et al. (2021b), which can be easily adjusted to various settings.

## B ADDITIONAL EXPERIMENTS

In this Appendix, we first explain further experimental details in Appendix B.1. Then we conduct more ablation studies about data destruction in Appendix B.2 and operations of data destruction in Appendix B.3. Next we experiment with top-k confidence voting in Appendix B.4. This is a generalization of FedOV where we just count k most confident votes. For heavier models and more complicated datasets, we experiment with VGG-9 in Appendix B.5 and ResNet-50 on CIFAR-100 in Appendix B.6. In Appendix B.7, we thoroughly compare our data destruction with a similar method Cutpaste Li et al. (2021a). We justify that data destruction can generate more diverse outliers and achieve higher accuracy. More experiments on scalability are conducted in Appendix B.8. FedOV is further compared with FedDF under different label skews (Appendix B.9), and with FedAwS under #C=1 partition (Appendix B.10). We also evaluate the effectiveness of DD and AOE on centralized OSR settings in Appendix B.11. Finally, we compare FedOV with baselines under multiple rounds in Appendix B.12.

### B.1 FURTHER DETAILS

We summarize the datasets in our experiments in Table 6.

Table 6: Basic information of datasets we use.

| Datasets | Training sample size | Test sample size | Input dimension | # of classes |
|----------|---------------------|------------------|-----------------|--------------|
| CIFAR-10 | 50,000 | 10,000 | 3,072 | 10 |
| SVHN | 73,257 | 26,032 | 3,072 | 10 |
| FMNIST | 60,000 | 10,000 | 784 | 10 |
| MNIST | 60,000 | 10,000 | 784 | 10 |
| CIFAR-100 | 50,000 | 10,000 | 3,072 | 100 |

For PROSER, we choose $\beta = 0.01, \gamma = 1$, according to the default trade-off parameter setting in the official code[1]. For ease of implementation, we omit the extension to multiple dummy classifiers.

For data destruction, details of our current transformations are as follow. The core idea is to destroy the key features of the original image while keeping some scrappy features.

- Random resized crop: with scale range (0.1, 0.33). We choose a small portion in order not to contain the key features and enlarge that portion to original size.

- Gaussian blur: with random kernel ranging from 1*3 to 5*9, and random $\sigma \in (10, 100)$. We choose a large $\sigma$ to blur out key features.

- Random erasing: with scale range (0.33, 0.5). We choose a large portion to erase in order to spoil key features.

---

[1] https://github.com/zhoudw-zdw/CVPR21-Proser

- Random paste: We random paste half of the image to another place.

- Random swap: We swap left-side and right-side, or upper-side and down-side.

- Random rotation: We random rotate two square portions of the image.

For adversarial learning, we set 5 local steps and each step size 0.002.

For Distilled FedOV, the local training step is the same as FedOV. After the server collects all local models, it performs knowledge distillation based on the open-set voting results. Formally, denote the local models $f_1, ..., f_N$, and each model outputs $c + 1$ scores where the last score is for the class "unknown". The student model $f_s$ outputs $c$ scores for the $c$ known classes.

For an input $x$, we add the output probability of each local model $scores(x) = \sum_{i=1}^{N} \sigma(f_i(x))$ where $\sigma$ is the softmax function. The voting result is the first $c$ scores for the $c$ known classes, i.e. $v(x) = scores(x)_{0,1,...,c-1}$. The normalized probability $v_n(x) = \frac{v(x)}{|v(x)|_1}$ is used as the target to distill the student model. Therefore, we have distillation loss $L_{dis} = KL(\sigma(f_s(x)), v_n(x))$, where $KL$ is Kullback–Leibler divergence.

By default, we use the first half of the test set as the public unlabelled dataset for knowledge distillation in the server and then test the distilled model on the second half of the test set. We use Adam optimizer with learning rate 0.001, and train 100 epochs on the public unlabelled dataset for the distillation process.

For FedProx, we tune the hyper-parameter $\mu \in \{0.001, 0.01, 0.1, 1\}$. For methods requiring knowledge distillation, we use first half of test set as public unlabelled dataset. The other half are used to compute accuracy.

Our simple CNN contains two 5*5 convolution layers with 2*2 max pooling layer. The first has 6 channels and the second has 16 channels. Then it has two fully-connected layers with 120 and 84 neurons separately. We use ReLu as the activation function between layers.

All experiments are conducted on a single 3090 GPU. We compare computing time of different algorithms in Table 7.

Table 7: Running time of different algorithms with simple CNN on CIFAR-10 dataset. There are 10 clients and each client runs 200 local epochs with only one communication. Our device is a single 3090 GPU.

| FedOV | Close-set voting | FedAvg | FedProx | FedNova |
|---|---|---|---|---|
| ∼3.0 h | ∼1.5 h | ∼1.5 h | ∼2.0 h | ∼1.5 h |

For the additional time cost of using data destruction (DD) and adversarial outlier enhancement (AOE), it costs about 2 times computation during local training in our experiments, depending on different datasets. The experiments shown in Table 8 are with partition $p_k \sim Dir(0.5)$ into 10 clients.

Table 8: Time for the first client to finish local training of 200 epochs. Our device is a single 3090 GPU.

| Dataset | FedOV | Open-set voting w/o DD & AOE |
|---|---|---|
| CIFAR-10 | $\sim 17$ min | $\sim 9$ min |
| FMNIST | $\sim 12$ min | $\sim 6$ min |

For more implementation details, please refer to our source code.

## B.2 USING ADVERSARIAL LEARNING WITHOUT DATA DESTRUCTION

In this section, we test on the effect of adversarial learning. Adversarial samples are based on outliers generated by data destruction. Here we also use data destruction outliers to generate adversarial samples, but we omit the loss of data destruction (DD). Results are shown in Table 9. Generally, omitting the loss of DD is not a good choice.

Table 9: Experimental results of different FL voting strategies with simple CNN model. We repeat all experiments for three times.

| Dataset | Partition | Open-set (PROSER + AOE) | FedOV |
|---|---|---|---|
| CIFAR-10 | $\#C = 1$ | 39.2%±3.1% | **40.0%±1.7%** |
| | $\#C = 2$ | 35.8%±3.0% | **42.0%±2.4%** |
| | $\#C = 3$ | **57.6±0.3%** | 55.6%±6.3% |
| | $p_k \sim Dir(0.5)$ | **65.7%±0.7%** | **65.7%±0.7%** |
| | $p_k \sim Dir(0.1)$ | 61.3%±0.4% | **61.7%±1.1%** |

## B.3 COMPARING WITH USING SINGLE TRANSFORMATION

In this section, we verify the effectiveness of random data destruction from a pool of transformations. We compare with using single transformation. We experiment on CIFAR-10 dataset, and all algorithms are open-set voting containing the same PROSER loss. The differences are their data destruction strategies. Results are shown in Table 10. None of single transformation can reach the accuracy of random transformation from data destruction set.

Table 10: Experimental results of using random data destruction from a pool of transformations, compared with using single transformation.

| Partition | Data destruction | Random resized crop | Gaussian blur | Random erasing | Random paste | Random swap | Random rotation |
|---|---|---|---|---|---|---|---|
| $\#C = 1$ | **33.5%±2.3%** | 15.2%±0.8% | 16.1%±0.7% | 11.6%±1.6% | 23.9%±1.5% | 32.3%±0.3% | 25.6%±0.8% |
| $\#C = 2$ | 41.3%±7.7% | 37.7%±0.5% | 38.6%±2.0% | 36.4%±1.6% | **42.4%±5.0%** | 37.3%±5.6% | 35.5%±7.1% |
| $\#C = 3$ | **54.3%±2.1%** | 50.5%±3.6% | 52.3%±5.3% | 48.7%±6.0% | 52.3%±4.0% | 53.8%±2.8% | 51.2%±4.0% |
| $p_k \sim Dir(0.5)$ | 67.6%±0.3% | 66.0%±1.1% | 66.4%±1.2% | 66.5%±0.8% | 66.6%±0.7% | 67.8%±0.8% | **68.0%±0.9%** |
| $p_k \sim Dir(0.1)$ | **61.3%±1.0%** | 56.9%±3.6% | 56.1%±2.7% | 54.9%±2.3% | 56.4%±2.6% | 58.2%±2.9% | 57.9%±2.0% |

## B.4 TOP-k CONFIDENCE VOTING

A possible alteration to our framework is to select the top k confident models for voting. Specifically, after getting predictions of all models, the server sorts all predictions by the probability of "unknown" channel. Then we sum up predictions of the lowest k "unknown" (i.e. the top k confident) while discarding all other predictions. Hyper-parameter k can be tuned for different tasks.

We visualize the voting accuracy curve versus k for experiments of VGG-9 on CIFAR-10 in Figure 5. As we can see, the best k differs for different label skews. For extreme $\#C = 2$, $k = 1$ seems the best. For slight skew $p_k \sim Dir(0.5)$, $k = 10$ is the best. Under $\#C = 2$ cases, each model is trained only on two classes, in such case we would better follow the most confident expert, since others are more likely to make wrong prediction. Under $p_k \sim Dir(0.5)$, models are trained on more classes and are generally more clever, where considering all the models for prediction becomes better.

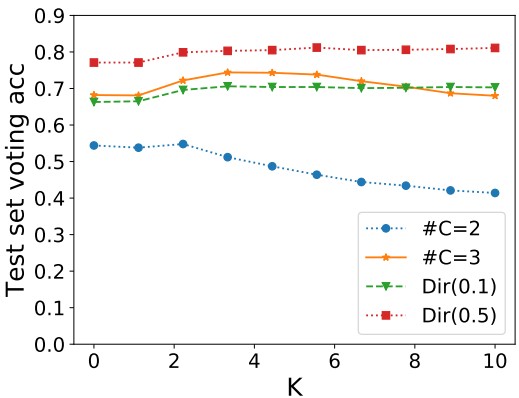

Figure 5: Voting accuracy on different k of VGG-9 on CIFAR-10.

By tuning k for different settings, results of best top-k confidence voting is shown in Table 11. For close-set voting, we report the accuracy of counting votes from all 10 clients, since all clients are equal. For open-set voting, we report the best accuracy among counting votes from top 1, 2, 3, ..., 10 confident clients. We can get similar conclusions as in Section 4.3.

Table 11: Experimental results of different FL voting strategies with simple CNN model. We report the best top-k confidence voting accuracy. We repeat all experiments for three times.

| Dataset | Partition | Close-set | Open-set (PROSER) | Open-set (PROSER + DD) | FedOV |
|---|---|---|---|---|---|
| CIFAR-10 | $\#C=1$ | 10.2%±0.2% | 10.9%±0.6% | 33.5%±2.3% | **40.0%±1.7%** |
| | $\#C=2$ | 37.2%±2.5% | 43.9%±0.3% | **49.7%±0.5%** | 48.9%±1.2% |
| | $\#C=3$ | 43.2%±2.7% | 54.9%±2.0% | **58.9%±0.6%** | 58.8%±1.6% |
| | $p_k \sim Dir(0.5)$ | 65.0%±0.1% | 66.7%±0.1% | **67.6%±0.3%** | 65.7%±0.7% |
| | $p_k \sim Dir(0.1)$ | 55.9%±1.3% | 58.0%±0.8% | 61.3%±1.0% | **61.7%±1.1%** |
| SVHN | $\#C=1$ | 7.3%±0.1% | 6.8%±0.1% | 47.3%±1.3% | **64.5%±1.9%** |
| | $\#C=2$ | 35.7%±7.6% | 60.2%±2.2% | 62.3%±1.7% | **74.7%±0.9%** |
| | $\#C=3$ | 60.1%±9.5% | 73.1%±2.9% | 73.6%±1.1% | **81.4%±1.2%** |
| | $p_k \sim Dir(0.5)$ | 85.1%±0.4% | 85.4%±0.3% | 84.9%±0.3% | **85.7%±0.3%** |
| | $p_k \sim Dir(0.1)$ | 64.9%±0.4% | 69.2%±1.1% | 74.6%±0.9% | **78.7%±0.6%** |
| FMNIST | $\#C=1$ | 10.1%±0.5% | 18.8%±1.4% | 71.0%±2.1% | **73.3%±1.5%** |
| | $\#C=2$ | 36.3%±5.5% | 46.2%±2.7% | **70.5%±3.6%** | 74.9%±2.2% |
| | $\#C=3$ | 57.0%±7.0% | 64.4%±0.4% | 76.0%±2.8% | **77.4%±1.6%** |
| | $p_k \sim Dir(0.5)$ | 88.5%±0.2% | 88.7%±0.2% | **89.1%±0.1%** | 89.0%±0.3% |
| | $p_k \sim Dir(0.1)$ | 73.6%±0.4% | 73.2%±0.9% | 77.9%±0.6% | **78.5%±0.6%** |
| MNIST | $\#C=1$ | 15.5%±2.8% | 17.3%±0.6% | 76.6%±8.3% | **79.4%±1.7%** |
| | $\#C=2$ | 44.3%±6.4% | 69.3%±2.4% | **88.6%±1.4%** | 88.2%±4.8% |
| | $\#C=3$ | 59.6%±7.0% | 87.3%±3.9% | 93.7%±2.4% | **96.2%±1.1%** |
| | $p_k \sim Dir(0.5)$ | 98.3%±0.1% | 98.3%±0.1% | 98.5%±0.1% | **98.6%±0.1%** |
| | $p_k \sim Dir(0.1)$ | 93.3%±0.4% | 95.4%±0.2% | **96.6%±0.1%** | **96.6%±0.1%** |

## B.5 USING HEAVIER MODEL

In this section, we use heavier VGG-9 model for CIFAR-10, since CIFAR-10 underfits in a simple CNN model. Results are shown in Table 12 and 13. For VGG-9 experiments, our method still works.

Table 12: Experimental results of different FL voting strategies with VGG-9 model. We repeat all experiments for three times.

| Dataset | Partition | Close-set | Open-set (PROSER) | Open-set (PROSER + DD) | FedOV |
|---|---|---|---|---|---|
| CIFAR-10 | $\#C=1$ | 10.2%±0.3% | 13.2%±0.6% | 32.7%±4.2% | **46.4%±1.3%** |
| | $\#C=2$ | 36.2%±2.4% | 35.1%±4.8% | 38.6%±10.3% | **42.0%±1.4%** |
| | $\#C=3$ | 50.8%±11.0% | 51.1%±1.8% | 62.4%±6.7% | **63.0%±4.4%** |
| | $p_k \sim Dir(0.5)$ | 75.8%±4.7% | 79.0%±0.5% | **80.7%±0.4%** | 80.6%±0.5% |
| | $p_k \sim Dir(0.1)$ | 60.9%±1.5% | 63.7%±0.7% | 68.3%±0.4% | **70.6%±0.4%** |

Table 13: Experimental results of different FL voting strategies with VGG-9 model. We report the best top-k confidence voting accuracy. We repeat all experiments for three times.

| Dataset | Partition | Close-set | Open-set (PROSER) | Open-set (PROSER + DD) | FedOV |
|---|---|---|---|---|---|
| CIFAR-10 | $\#C=1$ | 10.2%±0.3% | 13.2%±0.6% | 33.1%±4.4% | **47.1%±1.5%** |
| | $\#C=2$ | 36.2%±2.4% | 42.7%±4.5% | 56.0%±2.1% | **59.0%±4.0%** |
| | $\#C=3$ | 50.8%±11.0% | 64.2%±4.4% | 66.3%±4.1% | **71.9%±2.9%** |
| | $p_k \sim Dir(0.5)$ | 75.8%±4.7% | 79.1%±0.6% | **80.8%±0.4%** | **80.8%±0.4%** |
| | $p_k \sim Dir(0.1)$ | 60.9%±1.5% | 67.8%±0.8% | 71.0%±0.5% | **72.3%±1.4%** |

### B.6 EXPERIMENTS ON CIFAR-100 WITH RESNET-50

Besides VGG-9, we also experiment on ResNet-50 to verify the effectiveness of FedOV. We use CIFAR-100 to test on more complicated datasets. Results are shown in Table 14. For the baselines of one-round FedAvg, FedProx, SCAFFOLD and FedNova, we omit them since their one-shot accuracy are much lower (see Table 1, 4, 17). FedKT trains multiple models in each client and has huge computation and storage cost when using ResNet-50, so we omit it. Actually in FedKT paper (Li et al., 2021c), the authors do not experiment with heavy models like ResNet-50 either. Note that ResNet-50 has batch normalization layers. Therefore in each training batch, we have to mix train data and generated outliers. Previously we compute a batch of train data and another batch of generated outliers separately. For ResNet-50, it can cause problem due to batch normalization. To speed up computation, we only use data destruction strategy in our experiments for FedOV. Experimental results show that FedOV is better than close-set voting and FedDF for CIFAR-100 with ResNet-50 model.

Table 14: Experimental results on ResNet-50. We run 100 local epochs each client, and train the student model for 100 epochs during distillation.

| Dataset | Client Number | Partition | FedOV | Distilled FedOV | Close-set | Distilled close-set | FedDF |
|---|---|---|---|---|---|---|---|
| CIFAR-100 | 10 | $p_k \sim Dir(0.5)$ | **54.0%** | 31.7% | 51.6% | 30.7% | 27.5% |
| | | $p_k \sim Dir(0.1)$ | **47.3%** | 26.6% | 44.7% | 25.6% | 22.5% |
| | 100 | $\#C = 1$ | **3.6%** | 2.2% | 1.0% | 1.0% | 0.6% |
| | | $\#C = 2$ | **9.4%** | 5.9% | 4.8% | 4.9% | 1.2% |
| | | $\#C = 3$ | **13.4%** | 9.9% | 10.0% | 8.9% | 1.6% |

### B.7 COMPARISON WITH CUTPASTE

We find that an existing study proposed CutPaste (Li et al., 2021a) for outlier detection task, which generates outliers by applying operations on the training data. However, CutPaste contains limited operations and these operations cannot effectively corrupt the original features.

In this section, we compare our data destruction with Cutpaste (Li et al., 2021a) and explains why we do not include Cutpaste in our main experiments.

First, we show the outliers generated by Cutpaste in Figure 6. Cutpaste outliers are less diverse than our data destruction in Figure 3a.

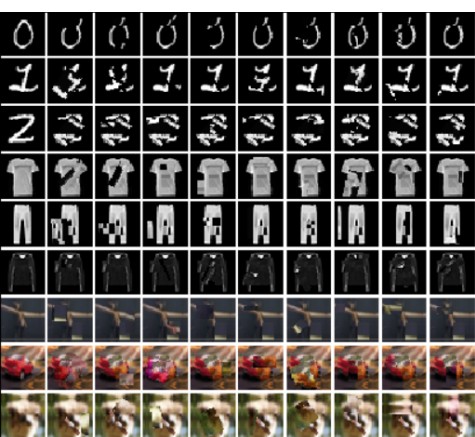

Figure 6: Outliers generated by Cutpaste. The first column is real samples and the other columns are Cutpaste outliers.

Next, we show some experimental results. We experiment on simple CNN model under CIFAR-10 setting. Mixture means for each image, we augment using our method and Cutpaste each with 50% chance. Results are shown in Table 15. Even in some cases, Cutpaste or mixture can slightly

outperform our method, in other cases our method can greatly outperform them. Therefore, adding Cutpaste can hardly bring significant improvement.

Table 15: Experimental results of Cutpaste and our augmentation. We report the accuracy of counting votes from all 10 clients.

| Dataset | Partition | Open-set (PROSER + Cutpaste) | Open-set (PROSER + mixture) | Open-set (PROSER + DD) |
|---|---|---|---|---|
| CIFAR-10 | $\#C = 1$ | 17.8%±2.2% | 24.7%±1.7% | **33.5%±2.3%** |
| | $\#C = 2$ | 36.7%±2.0% | 39.0%±3.8% | **41.3%±7.7%** |
| | $\#C = 3$ | 51.5%±3.2% | 52.8%±2.4% | **54.3%±2.1%** |
| | $p_k \sim Dir(0.5)$ | 67.7%±0.7% | **68.5%±0.5%** | 67.6%±0.3% |
| | $p_k \sim Dir(0.1)$ | 57.5%±2.2% | 59.4%±1.7% | **61.3%±1.0%** |

We also conduct similar experiments based on the metric of outlier detection. Our dataset includes MNIST, Fashion-MNIST and CIFAR-10. We use one class when training and all classes of the same dataset when testing. The metric is ROC-AUC of the outlier probability. We run 200 local epochs with batch size 64, learning rate 0.001. For ROC-AUC, we report the average of last 10 epochs. Results are shown in Table 16. Our augmentation method still outperforms Cutpaste and mixture.

In conclusion, our approach can generate more diverse outliers and achieve better accuracy compared with CutPaste.

Table 16: Experimental results of cutpaste and our augmentation under common outlier detection metric

| Dataset | Known class | Open-set (PROSER + Cutpaste) | Open-set (PROSER + mixture) | Open-set (PROSER + DD) |
|---|---|---|---|---|
| FMNIST | 0 | 84.7% | 87.3% | **93.7%** |
| | 1 | 96.9% | **98.1%** | **98.1%** |
| | 2 | 82.9% | 86.9% | **89.4%** |
| | 3 | 87.4% | 85.7% | **89.2%** |
| | 4 | 81.7% | 87.7% | **93.1%** |
| | 5 | 89.7% | 90.1% | **91.7%** |
| | 6 | 73.4% | 78.1% | **80.7%** |
| | 7 | 97.4% | **97.6%** | 95.8% |
| | 8 | 84.8% | 84.4% | **91.3%** |
| | 9 | 97.2% | **97.8%** | 96.8% |
| | Average | 87.6% | 89.4% | **92.0%** |
| MNIST | 0 | 97.9% | 98.6% | **99.1%** |
| | 1 | **96.4%** | 95.5% | 89.0% |
| | 2 | 84.7% | 83.2% | **97.8%** |
| | 3 | 82.3% | 83.1% | **94.1%** |
| | 4 | 90.6% | 92.8% | **94.4%** |
| | 5 | 88.2% | 86.4% | **95.7%** |
| | 6 | 96.1% | 98.6% | **98.9%** |
| | 7 | 86.8% | 92.5% | **92.8%** |
| | 8 | 83.1% | 90.0% | **96.7%** |
| | 9 | 93.2% | 95.1% | **98.1%** |
| | Average | 89.9% | 91.6% | **95.7%** |
| CIFAR-10 | 0 | 56.9% | 60.8% | **68.1%** |
| | 1 | 48.7% | 72.1% | **81.8%** |
| | 2 | 58.0% | 58.9% | **66.4%** |
| | 3 | 59.2% | **62.6%** | 61.2% |
| | 4 | 58.1% | 59.0% | **62.2%** |
| | 5 | 69.1% | **74.2%** | 72.2% |
| | 6 | 61.0% | 66.0% | **71.8%** |
| | 7 | 62.2% | 68.6% | **76.3%** |
| | 8 | 62.1% | 63.6% | **71.5%** |
| | 9 | 54.7% | 61.8% | **73.8%** |
| | Average | 59.0% | 64.8% | **70.5%** |

## B.8 More Experiments on Scalability

In this section, we show results of scalability experiments on SVHN, Fashion-MNIST and MNIST. Except for datasets, other experimental settings are the same as in main paper. Results are shown in Table 17. FedOV still works well for scalability on SVHN, Fashion-MNIST and MNIST. Distilled FedOV can also outperform distilled close-set voting, FedAvg, FedProx and FedNova with single communication.

Both FedDF and distilled close-set voting can outperform FedAvg. Note that the ensemble of FedDF is based on average logits, while distilled close-set voting uses average probability, i.e. the softmax of logits. After softmax, the probability is limited in the range of [0,1]. However in FedDF, logits can be arbitrary real number. If a stupid local model outputs some very bad logits, it can greatly influence the ensemble of FedDF. This can probability explain why FedDF has different performance compared with distilled close-set voting, although both approaches are similar. Especially when there are 80 clients where it is more likely to have some very stupid local models, FedDF has worse performance than distilled close-set voting.

Table 17: Experimental results of different number of clients with simple CNN model.

| Dataset | Client Number | Partition | FedOV | Distilled FedOV | Close-set | Distilled close-set | FedAvg | FedProx | FedNova | SCAFFOLD | FedDF | FedKT |
|---|---|---|---|---|---|---|---|---|---|---|---|---|
| SVHN | 20 | $\#C=1$ | **71.0%** | 67.1% | 16.1% | 16.4% | 11.1% | 8.4% | 6.1% | 8.8% | 19.7% | 6.7% |
| | | $\#C=2$ | **80.0%** | 76.2% | 33.4% | 31.9% | 25.3% | 10.2% | 10.3% | 9.8% | 45.9% | 45.9% |
| | | $\#C=3$ | **84.3%** | 80.6% | 56.1% | 52.4% | 24.1% | 19.4% | 20.1% | 14.2% | 60.3% | 58.8% |
| | | $p_k \sim Dir(0.5)$ | **86.1%** | 81.7% | 85.4% | 81.8% | 39.4% | 36.1% | 38.2% | 30.2% | 76.4% | 72.0% |
| | | $p_k \sim Dir(0.1)$ | **82.7%** | 79.2% | 74.0% | 71.4% | 22.6% | 24.6% | 23.6% | 20.0% | 66.3% | 65.0% |
| | 40 | $\#C=1$ | **77.6%** | 72.6% | 6.4% | 6.6% | 19.6% | 9.7% | 6.4% | 10.9% | 6.5% | 6.7% |
| | | $\#C=2$ | **81.6%** | 78.3% | 51.5% | 49.5% | 19.6% | 19.9% | 8.7% | 19.6% | 69.3% | 53.4% |
| | | $\#C=3$ | **85.4%** | 82.4% | 76.8% | 74.1% | 19.6% | 21.3% | 23.2% | 19.9% | 67.9% | 68.4% |
| | | $p_k \sim Dir(0.5)$ | **86.1%** | 83.0% | 84.7% | 81.2% | 21.1% | 21.4% | 23.3% | 19.6% | 76.9% | 67.3% |
| | | $p_k \sim Dir(0.1)$ | **81.2%** | 79.5% | 72.0% | 70.6% | 19.6% | 19.6% | 16.4% | 19.6% | 65.5% | 26.8% |
| | 80 | $\#C=1$ | **80.2%** | 79.0% | 7.3% | 7.2% | 19.6% | 18.6% | 6.4% | 19.3% | 6.6% | 6.7% |
| | | $\#C=2$ | **84.6%** | 82.1% | 77.8% | 75.0% | 22.9% | 19.7% | 6.4% | 19.6% | 69.6% | 42.0% |
| | | $\#C=3$ | **84.9%** | 81.4% | 81.0% | 79.7% | 21.7% | 23.6% | 20.7% | 19.6% | 62.8% | 46.6% |
| | | $p_k \sim Dir(0.5)$ | **84.5%** | 81.9% | 80.5% | 78.3% | 23.6% | 26.2% | 25.0% | 19.6% | 68.4% | 45.8% |
| | | $p_k \sim Dir(0.1)$ | **81.4%** | 77.9% | 68.5% | 66.9% | 19.6% | 19.6% | 23.9% | 19.6% | 55.8% | 18.0% |
| FMNIST | 20 | $\#C=1$ | **77.2%** | 74.9% | 10.7% | 10.6% | 10.0% | 12.8% | 10.0% | 10.0% | 10.0% | 9.9% |
| | | $\#C=2$ | **72.4%** | 69.0% | 41.9% | 39.6% | 10.0% | 10.0% | 10.0% | 10.0% | 35.9% | 34.8% |
| | | $\#C=3$ | **76.1%** | 73.3% | 58.2% | 54.8% | 37.2% | 32.2% | 36.0% | 33.1% | 47.9% | 48.7% |
| | | $p_k \sim Dir(0.5)$ | **87.5%** | 84.0% | 87.2% | 83.6% | 61.6% | 63.3% | 64.8% | 58.9% | 80.3% | 80.5% |
| | | $p_k \sim Dir(0.1)$ | **72.7%** | 70.2% | 66.7% | 65.6% | 40.1% | 36.0% | 35.8% | 34.4% | 72.3% | 68.0% |
| | 40 | $\#C=1$ | **75.7%** | 74.1% | 5.8% | 5.8% | 14.1% | 17.4% | 13.7% | 14.2% | 10.4% | 9.9% |
| | | $\#C=2$ | **77.6%** | 75.5% | 55.3% | 55.3% | 22.4% | 23.5% | 18.8% | 14.3% | 48.4% | 47.6% |
| | | $\#C=3$ | **68.7%** | 68.0% | 49.4% | 48.6% | 39.0% | 42.8% | 46.4% | 23.2% | 56.1% | 61.7% |
| | | $p_k \sim Dir(0.5)$ | **86.9%** | 82.6% | 86.2% | 83.9% | 58.4% | 60.4% | 58.5% | 42.0% | 79.9% | 77.0% |
| | | $p_k \sim Dir(0.1)$ | **81.3%** | 78.2% | 74.4% | 74.1% | 25.1% | 24.3% | 27.9% | 15.9% | 64.2% | 61.4% |
| | 80 | $\#C=1$ | **77.8%** | 76.6% | 10.0% | 9.9% | 10.0% | 11.0% | 10.0% | 13.3% | 10.2% | 9.9% |
| | | $\#C=2$ | **78.8%** | 75.2% | 54.4% | 52.8% | 35.6% | 35.5% | 35.7% | 18.2% | 47.4% | 69.8% |
| | | $\#C=3$ | **84.6%** | 81.6% | 66.2% | 63.5% | 51.5% | 45.6% | 47.7% | 31.9% | 59.0% | 73.9% |
| | | $p_k \sim Dir(0.5)$ | **85.3%** | 83.5% | 84.5% | 82.2% | 57.7% | 57.0% | 61.1% | 44.7% | 73.9% | 72.6% |
| | | $p_k \sim Dir(0.1)$ | **78.9%** | 76.8% | 71.7% | 69.9% | 32.4% | 38.2% | 35.9% | 18.0% | 63.7% | 68.2% |
| MNIST | 20 | $\#C=1$ | **88.5%** | 91.6% | 18.6% | 19.5% | 11.4% | 11.4% | 11.4% | 11.4% | 10.3% | 10.3% |
| | | $\#C=2$ | **86.1%** | 84.5% | 44.6% | 45.6% | 24.9% | 21.1% | 24.9% | 28.5% | 48.3% | 29.8% |
| | | $\#C=3$ | **88.9%** | 86.4% | 59.0% | 57.1% | 56.1% | 55.9% | 51.4% | 41.6% | 78.7% | 58.4% |
| | | $p_k \sim Dir(0.5)$ | **98.2%** | 97.8% | 97.3% | 97.2% | 77.9% | 74.4% | 78.4% | 66.7% | 97.5% | 95.5% |
| | | $p_k \sim Dir(0.1)$ | **87.5%** | 88.2% | 83.2% | 84.9% | 44.0% | 32.6% | 45.4% | 36.5% | 94.0% | 80.2% |
| | 40 | $\#C=1$ | **83.1%** | 84.7% | 10.4% | 11.5% | 13.5% | 15.3% | 9.5% | 10.5% | 10.3% | 10.3% |
| | | $\#C=2$ | **87.1%** | 87.6% | 60.9% | 62.0% | 42.1% | 33.3% | 32.0% | 20.2% | 34.5% | 33.0% |
| | | $\#C=3$ | **97.1%** | 97.5% | 69.6% | 68.7% | 60.2% | 54.9% | 50.1% | 44.5% | 80.9% | 64.1% |
| | | $p_k \sim Dir(0.5)$ | **97.6%** | 97.1% | 96.4% | 97.2% | 72.6% | 72.8% | 73.2% | 72.5% | 87.4% | 92.1% |
| | | $p_k \sim Dir(0.1)$ | **94.2%** | 93.6% | 85.1% | 86.1% | 38.0% | 49.9% | 43.9% | 42.6% | 84.6% | 79.5% |
| | 80 | $\#C=1$ | **89.5%** | 91.4% | 10.8% | 10.3% | 11.4% | 11.8% | 11.4% | 11.3% | 12.1% | 10.3% |
| | | $\#C=2$ | **96.0%** | 95.4% | 67.1% | 65.9% | 34.0% | 30.9% | 31.1% | 14.9% | 58.2% | 56.8% |
| | | $\#C=3$ | **96.9%** | 97.4% | 81.2% | 82.2% | 49.3% | 49.4% | 45.3% | 29.6% | 78.2% | 74.3% |
| | | $p_k \sim Dir(0.5)$ | **96.5%** | 96.9% | 94.4% | 95.6% | 69.7% | 70.5% | 71.2% | 59.0% | 81.6% | 85.6% |
| | | $p_k \sim Dir(0.1)$ | **94.2%** | 94.4% | 86.1% | 89.0% | 40.3% | 42.6% | 41.1% | 36.0% | 79.0% | 86.7% |

## B.9 Comparing Distilled FedOV with FedDF

We compare Distilled FedOV with FedDF under various label skews in Table 18. We have 10 clients and each client trains 100 local epochs. The model is simple CNN and we repeat all experiments three times with random seed 0,1,2 respectively. $p_k \sim Dir(2)$ is the most IID partition, while $p_k \sim Dir(0.1)$ is the most non-IID partition. As we can see, Distilled FedOV outperforms FedDF in all settings. In the more non-IID setting $p_k \sim Dir(0.1)$, the improvement of Distilled FedOV is significantly more than the other two settings.

Table 18: Comparison between distilled FedOV and FedDF under different levels of label skews.

| Partition | Dataset | Distilled FedOV | FedDF | Improvement |
|---|---|---|---|---|
| $p_k \sim Dir(2)$ | MNIST | **98.4%±0.3%** | 97.6%±0.5% | 0.8% |
| | SVHN | **84.7%±0.4%** | 82.7%±2.2% | 2.0% |
| | FMNIST | **85.6%±0.5%** | 81.8%±1.9% | 3.8% |
| | CIFAR-10 | **53.2%±0.6%** | 43.3%±1.6% | 9.9% |
| $p_k \sim Dir(0.5)$ | MNIST | **97.7%±0.3%** | 97.1%±0.3% | 0.6% |
| | SVHN | **81.8%±1.4%** | 80.6%±3.2% | 1.2% |
| | FMNIST | **85.0%±0.1%** | 80.6%±3.3% | 4.4% |
| | CIFAR-10 | **51.6%±1.4%** | 34.1%±1.9% | 17.5% |
| $p_k \sim Dir(0.1)$ | MNIST | **85.4%±1.7%** | 80.2%±6.7% | 5.2% |
| | SVHN | **73.8%±1.0%** | 61.7%±7.3% | 12.1% |
| | FMNIST | **73.4%±1.5%** | 60.3%±9.6% | 13.1% |
| | CIFAR-10 | **45.5%±0.9%** | 22.9%±4.3% | 22.6% |

## B.10  COMPARING WITH FEDAWS ON #C=1

For the setting where each client only has one class, FedAwS (Yu et al., 2020) proposes spreadout regularization to push the embeddings of each class apart from each other to avoid all inputs collapsing to a single point. FedUV (Hosseini et al., 2021) argues that FedAwS leaks the sensitive class embedding information to the server. The authors propose to use error-correcting codes to protect the embeddings and achieve similar accuracy compared with FedAwS. In FedUV paper (Hosseini et al., 2021), both FedUV and FedAwS are compared with FedAvg with regular softmax cross-entropy loss function on user verification tasks. The authors conclude that FedAvg with regular softmax cross-entropy loss achieves the best accuracy in most cases, however regular FedAvg leaks class embeddings to the server and all other clients. FedUV focuses more on privacy protection and has lower accuracy than FedAvg, while we focus on accuracy and our FedOV has significantly outperformed FedAvg, therefore we do not compare with FedUV.

There are also works specifically for the real-world context of #C=1, such as federated face recognition (Liu et al., 2022). However, it focuses on personalized FL, and each client can access a public dataset to assist training, which is different from our settings. Therefore, we do not compare with it.

Next, we compare FedOV with FedAwS on #C=1 setting. We use the default $k = 10$ in FedAwS paper. Since we only have 10 clients, it equals calculating the distance with all other class embeddings. We tune the best $\lambda \in \{10, 100\}$ according to the authors' suggestions. We use the same squared hinge loss with hyper-parameter 0.9 as the original paper. FedAwS paper does not state the number of local epochs. From our experiments, one local epoch is enough for convergence under #C=1. For successive epochs, the loss is almost zero and the accuracy does not improve. By default, for both FedAwS and FedAvg, we train 1 local epoch with learning rate 0.001.

Results are shown in Table 19. As we can see, FedOV significantly outperforms FedAwS and FedAvg in one round. After one round, FedAwS and FedAvg achieve similar accuracy.

Table 19: Comparison between FedOV, FedAwS, and FedAvg under #C=1 partition. The model is simple CNN. We run three times for each setting.

| Dataset | FedOV | FedAwS | FedAvg |
|---|---|---|---|
| MNIST | **79.3%±1.8%** | 11.6%±2.3% | 10.1%±1.3% |
| SVHN | **64.5%±1.9%** | 19.6%±0.0% | 18.2%±1.5% |
| FMNIST | **73.3%±1.6%** | 10.4%±0.7% | 10.2%±0.3% |
| CIFAR-10 | **40.0%±1.7%** | 10.0%±0.4% | 11.3%±1.4% |

## B.11  PERFORMANCE OF DD AND AOE ON CENTRALIZED OSR EXPERIMENTAL SETTINGS

Our two techniques DD and AOE can also be extended to centralized OSR training to augment the outliers. In this section, we explore how they perform in centralized OSR experimental settings. It clarifies why DD and AOE are more suitable and more important for FL label skew settings.

We utilize the common setting of OSR algorithm evaluation. For MNIST, Fashion-MNIST, CIFAR-10 and SVHN datasets, we regard the first 2, 4, 6 and 8 classes as known during training. During testing, all 10 classes in the test set appear.

We compare PROSER+DD+AOE with PROSER, and run 200 epochs each. The model is simple CNN. We show the results in Table 20. As we can see, under the OSR settings, adding our DD and AOE techniques can also bring improvement compared to the state-of-the-art PROSER algorithm in most cases. When there are fewer known classes, the average improvement brought by our techniques is larger. For the experimental settings of many OSR papers (Zhou et al., 2021; Neal et al., 2018; Perera et al., 2020; Yoshihashi et al., 2019), the OSR model is trained on at least 4 known classes. The scenario of a limited number of classes (such as 2 classes) is not studied.

For PROSER, we can see a clear trend that when the number of known classes decreases, the accuracy decreases. Since PROSER generates outliers between classes, when the number of known classes is limited, the generated outliers are not diverse enough to train a good OSR model. Therefore, PROSER is inapplicable to scenarios with limited known classes. Under FL label skews, some parties may have limited data for some labels due to privacy regulations and data distribution heterogeneity. Therefore, some clients may have limited known classes, e.g. only 1 class or 2 classes. And since there are multiple clients, it is unlikely to ensure that all clients have at least some number of classes. Such scenarios are different from typical centralized OSR experimental settings and pose great challenges for PROSER as we also show in Figure 2. Our two techniques DD and AOE can tackle these challenges under the scenario of a limited number of classes. There is no clear relation between the accuracy of adding our two techniques and the number of known classes. It is because the task difficulty is not linear w.r.t the number of known classes. More known classes can be both a benefit and a challenge. The benefit is that more examples of diverse classes help generate more diverse outliers in-between for better outlier detection. The challenge is that the task is more complicated as the model has to classify more seen classes.

In conclusion, although DD and AOE can be extended to centralized OSR settings, they can bring more significant improvement to FL and are more essential for FL with label skews, compared with PROSER when the number of classes is limited, such as in FL label skew scenarios.

Table 20: Test accuracy comparison between PROSER+DD+AOE with PROSER under centralized OSR settings. "Unknown" is regarded as a new class besides the seen classes. We run three times for each setting.

| # of known classes | Dataset | PROSER +DD+AOE | PROSER | Improvement | Average Improvement |
|---|---|---|---|---|---|
| 2 | MNIST | **84.2%±2.2%** | 51.1%±11.0% | 33.1% | 39.7% |
| | SVHN | **75.9%±0.9%** | 30.7%±2.8% | 45.2% | |
| | FMNIST | **70.1%±0.5%** | 25.2%±1.0% | 44.9% | |
| | CIFAR-10 | **60.2%±4.0%** | 24.6%±1.2% | 35.6% | |
| 4 | MNIST | **65.3%±1.8%** | 52.1%±5.8% | 13.2% | 15.9% |
| | SVHN | **68.4%±2.4%** | 56.4%±0.5% | 12.0% | |
| | FMNIST | **76.3%±0.4%** | 50.2%±1.5% | 26.1% | |
| | CIFAR-10 | **57.6%±2.1%** | 45.5%±0.6% | 12.1% | |
| 6 | MNIST | **70.9%±1.1%** | 65.9%±1.3% | 5.0% | 3.8% |
| | SVHN | 69.6%±1.3% | **70.2%±0.9%** | -0.6% | |
| | FMNIST | **68.4%±2.2%** | 58.3%±0.6% | 10.1% | |
| | CIFAR-10 | **54.4%±0.7%** | 53.7%±0.3% | 0.7% | |
| 8 | MNIST | **85.0%±1.0%** | 82.3%±1.2% | 2.7% | -2.0% |
| | SVHN | 70.2%±1.2% | **78.6%±0.6%** | -8.4% | |
| | FMNIST | **77.5%±0.2%** | 71.3%±0.5% | 6.2% | |
| | CIFAR-10 | 44.9%±2.3% | **53.3%±1.1%** | -8.4% | |

## B.12 COMPARING WITH BASELINES UNDER MULTIPLE ROUNDS

In this section, we run FedAvg, FedProx, SCAFFOLD, and FedNova for 200 communication rounds. There are 10 clients and each client trains 10 local epochs in each round. We compare FedOV with multi-round accuracy of baseline algorithms in Figure 7. As we can see, FedOV ensemble can achieve

the accuracy of baseline algorithms at 50-75 communication rounds. In extreme label skew #C=1, the four baseline FL algorithms cannot beat FedOV even after 200 communication rounds.

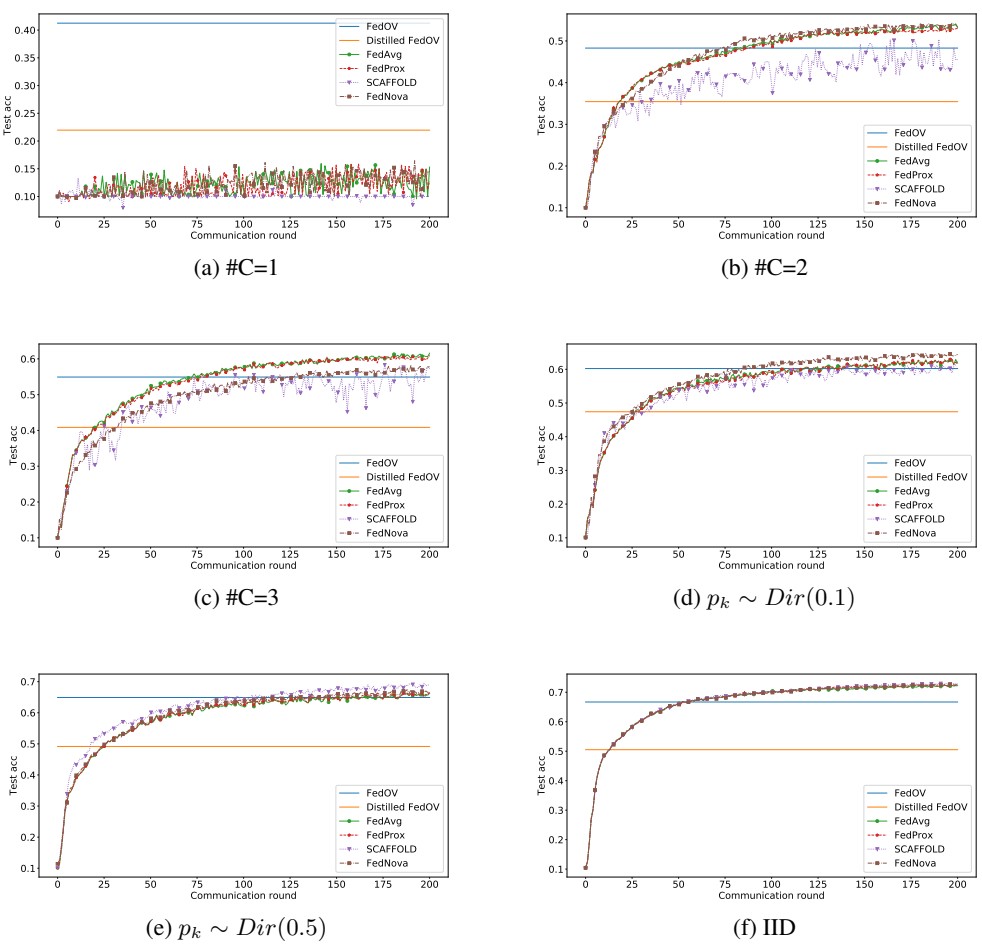

Figure 7: Comparing FedOV (one-shot FL accuracy) with baseline algorithms running multiple rounds. We experiment on six different partitions of CIFAR-10 dataset.

