# OpenReview forum: "Towards Addressing Label Skews in One-Shot Federated Learning"
_ICLR.cc/2023/Conference — ICLR 2023 poster_

### Official Review · Reviewer_Y1Mp · 2022-10-18

**Confidence:** 3
**Clarity, Quality, Novelty And Reproducibility:** The clarity, quality, novelty and rep…
**Correctness:** 3
**Technical Novelty And Significance:** 3
**Empirical Novelty And Significance:** 3
**Recommendation:** 8

**Strength And Weaknesses:**

Pros:

- The paper is technically sound and easy to understand.

- The method of introducing unknown class is novel and reasonable in FL learning.

- The experimental results show the effectiveness of the proposed method.

Cons:

- Lack of details of the experimental setting of distilled FedOV.

**Summary Of The Paper:**

This paper propose a new approach for one-shot federated learning, by generating diverse outliers and introduces them as an additional unknown class in local training to improve the voting performance. The paper proposes novel outlier generation approaches by corrupting the original features and further develop adversarial learning to enhance the outliers. The experimental results demonstrate the effectiveness of the proposed method.

**Summary Of The Review:**

In short, the paper is novel and technically sound and easy to understand. The experimental results demonstrate the effectiveness of this paper.

---

> ### Author Response · Authors · 2022-11-13
> **Response to reviewer Y1Mp**
>
> Thanks a lot for your constructive comments!
>
> >Q6. Lack of details of the experimental setting of distilled FedOV.
>
> We have added more details in Appendix B.1 as follows.
>
> For Distilled FedOV, the local training step is the same as FedOV. After the server collects all local models, it performs knowledge distillation based on the open-set voting results. Formally, denote the local models $f_1,...,f_N$, and each model outputs $c+1$ scores where the last score is for the class ``unknown''. The student model $f_s$ outputs $c$ scores for the $c$ known classes.
>
> For an input $x$, we add the output probability of each local model $scores(x)=\sum_{i=1}^N \sigma(f_i(x))$ where $\sigma$ is the softmax function. The voting result is the first $c$ scores for the $c$ known classes, i.e. $v(x)=scores(x)\_{0,1,...,c-1}$.
> The normalized probability $v_n(x)=\frac{v(x)}{|v(x)|\_1}$ is used as the target to distill the student model. Therefore, we have distillation loss $L_{dis}=KL(\sigma(f_s(x)), v_n(x))$, where $KL$ is Kullback–Leibler divergence.
>
> By default, we use the first half of the test set as the public unlabelled dataset for knowledge distillation in the server and then test the distilled model on the second half of the test set. We use Adam optimizer with learning rate 0.001, and train 100 epochs on the public unlabelled dataset for the distillation process.
>
> We have provided the code in the supplementary material for reproducibility.

---

> ### Author Response · Authors · 2022-11-22
> **Looking forward to your feedbacks for our rebuttal**
>
> Dear Reviewer Y1Mp,
>
> We have addressed your concerns and revised the paper. We'd appreciate it if you could provide any feedback and raise the score if your questions are solved. Thanks for your time and effort!

---

### Official Review · Reviewer_hyjU · 2022-10-19

**Confidence:** 3
**Correctness:** 3
**Technical Novelty And Significance:** 3
**Empirical Novelty And Significance:** 3
**Recommendation:** 6

**Clarity, Quality, Novelty And Reproducibility:**

Clarity: The paper is well-organized and clearly written.

Quality: Technically solid paper, with a high impact on the research field of federated learning.

Novelty: The paper makes non-trivial advances over the current state-of-the-art.

Reproducibility:  Key details are sufficiently well-described for competent researchers to confidently reproduce the main results.

**Strength And Weaknesses:**

### Strength

- The problem studied in this paper is interesting and valuable. Specifically, label skews is an essential issue that exists in one-shot federated learning and is urgent to be solved.
- This paper provides some novel perspectives. Applying Open-Set Recognition (OSR) into FL to introduce an unknown class to improve the voting is a novel view for overcoming label skews in one-shot FL.
- This paper conducts extensive experiments and demonstrates the advantages of the proposed method over the state-of-the-art LDL methods, which can effectively reflect the performance of the algorithm.
- The paper is well-organized and clearly written, which is easy to follow.
- The code is provided in the attachment for reproduction.

### Weaknesses

- In section 1, the author mentioned, "For example, in an extreme case where each client only has one label (e.g., face recognition), all clients predict the input as its own label and the voting result is meaningless". This case is a typical situation studied in this paper. However, there exist some works handling this case that are not mentioned in this paper, such as [1]. I think it is necessary to discuss and compare the proposed algorithm with these methods.

- In Table 3, one can notice that the performance of Distilled FedOV is worse than FedDF. The authors should analyze the reason for this phenomenon.

[1] Yu, Felix, et al. "Federated learning with only positive labels." *International Conference on Machine Learning*. 2020.

**Summary Of The Paper:**

This paper proposes a novel approach named FedOV which generates diverse outliers and introduces them as an additional unknown class in local training to address the label skew issue in one-shot Federated Learning (FL). Concretely, it proposes novel outlier generation approaches by corrupting the original features and further develops adversarial learning to enhance the outliers based on open-set recognition.

The main contributions of this paper are:

- This paper points out that the voting-based FL approaches are either directly used as a final model for predictions or distilled as a single model, thus failing to produce high-quality federated learning models.

- This paper may be the first time to propose open-set voting in FL by introducing the “unknown” class, which significantly improves the accuracy compared to traditional close-set voting in FL.
- This paper proposes two techniques (i.e., data destruction and adversarial outlier enhancement) for generating diverse “unknown” outliers without requirement on the number of classes of the training data.



**Summary Of The Review:**

I believe this paper is innovative and puts forward some new perspectives in the field of federated learning. However, I have concerns about whether this work can be accepted in its current form. I will update my reviews based on the authors' responses.

---

> ### Author Response · Authors · 2022-11-13
> **Response to reviewer hyjU**
>
> Thanks a lot for your constructive comments!
>
> >Q4. In section 1, the author mentioned, "For example, in an extreme case where each client only has one label (e.g., face recognition), all clients predict the input as its own label and the voting result is meaningless". This case is a typical situation studied in this paper. However, there exist some works handling this case that are not mentioned in this paper, such as [1]. I think it is necessary to discuss and compare the proposed algorithm with these methods.
>
> For the scenario where each client only has one label, we implement and compare with FedAwS [1] under #C=1 setting in Appendix B.10, Table 19. FedOV can still significantly outperform FedAwS, while FedAwS has a similar performance with FedAvg in one round.
>
> Besides, we also discuss some other works [2,3] under the case where each client only has one label. FedUV [2] focuses on privacy protection (protecting sensitive class embeddings) and experiments on user verification settings. According to experiments in [2], both FedUV and FedAwS have lower performance than pure FedAvg (with softmax cross entropy loss). However, FedAvg leaks class embeddings to the server and all clients. FedAwS leaks to the server, while FedUV can protect class embeddings. Since we focus on accuracy and already compare with FedAwS & FedAvg, we do not compare with FedUV. FedFR [3] explores a more specific face recognition context. It is personalized FL and distributes a public dataset to all clients to assist training, which is different from our settings, so we do not compare with it.
>
> [1] Yu, Felix, et al. "Federated learning with only positive labels." ICML 2020.
>
> [2] Hosseini, Hossein, et al. "Federated Learning of User Verification Models Without Sharing Embeddings." ICML 2021.
>
> [3] Liu, Chih-Ting, et al. "Fedfr: Joint optimization federated framework for generic and personalized face recognition." AAAI 2022
>
> >Q5. In Table 3, one can notice that the performance of Distilled FedOV is worse than FedDF. The authors should analyze the reason for this phenomenon.
>
> In the original draft, we run Distilled FedOV 3 times with different seeds. FedDF is run only once. To answer this question, we further run FedDF three times with the same three seeds. Combining these experiments, Distilled FedOV can outperform FedDF in all four datasets (Table 3).
>
> FedDF distillation is based on the average logits of local models, which is quite similar to Close-set Voting. The Dirichlet partition Dir(0.5) is a relatively slight non-IID partition, where Close-set Voting, as well as FedDF, have slight accuracy degradation. Therefore, in Dir(0.5) partition, the improvement of FedOV is relatively slight, compared to more non-IID cases. It can be seen from the scalability part (Table 4 & 17).
>
> We conduct additional experiments to compare Distilled FedOV and FedDF on Dir(2) & Dir(0.1) to justify this conclusion (Table 18). Dir(0.1) is the most non-IID case and Dir(2) is the most IID case. **Distilled FedOV can outperform FedDF in all settings.** The improvement of Distilled FedOV is larger compared with FedDF when the federated setting is more unbalanced, which further verifies the effectiveness of our techniques for non-IID data.

---

> ### Author Response · Authors · 2022-11-22
> **Looking forward to your feedbacks for our rebuttal**
>
> Dear Reviewer hyjU,
>
> We have addressed your concerns and revised the paper. We'd appreciate it if you could provide any feedback and raise the score if your questions are solved. Thanks for your time and effort!

---

### Official Review · Reviewer_ZcP3 · 2022-10-23

**Confidence:** 5
**Clarity, Quality, Novelty And Reproducibility:** Marginal
**Correctness:** 3
**Technical Novelty And Significance:** 2
**Empirical Novelty And Significance:** Not applicable
**Recommendation:** 6

**Strength And Weaknesses:**

Pros:
+ It is intriguing to introduce the concept of unknown class into federated learning and investigate a novel problem open set voting federated learning
+ The proposed method achieves new SOTA results on widely used benchmarks, which demonstrates the effectiveness of the proposed method.

Cons:
- The authors propose data destruction and adversarial outlier enhancement modules to respectively generate outliers from true samples and learn a tighter boundary. However, these two components seem not specifically aims for missing class issue in label skew one-shot fl. It would be better to further consider the relation with federated learning setting.

- The compared methods such as fedavg, fedprox, fednova are not designed for one-shot fl, it is nature that these methods perform worse on these experiments (Table1). So, it is not sure whether it is fair comparision.

- Typing Errors

1) ‘are used as part’  ‘are used as the part’\
2) ‘the above method’ ‘the above methods’
3) ‘train 100 epochs each client’ ‘train 100 epochs for each client’
4) ‘which causes massive’ ‘which cause massive’


**Summary Of The Paper:**

This paper focuses on the label skew problem in one-shot federated learning. The author identifies the problem is attributed to the limited scale of certain class in different participants, which brings unreliable voting scores and thus results in ineffective global model. In this paper, authors propose FedOV, which leverage generated diverse outliers as additional unknown class to increase voting efficiency. Extensive experiments are conducted on MNIST, Fashion-MNIST, CIFAR-10 and SVHN datasets.

**Summary Of The Review:**

This paper investigates an interesting problem, label skew in one-shot federated learning. But the proposed method is lack of novelty, seems the directly combines of existing strategies. Besides, authors do not provide in-depth rationale analysis and spend a large space of article to incorporate methodology into existing methods.

Post-rebuttal:
The authors provide comprehensive analysis in the response. Some issues have been clarified. I would like to raise my score to 6. However, some unclear parts should be clarified in the final version (if accepted).

---

> ### Author Response · Authors · 2022-11-13
> **Response to reviewer ZcP3 (1/2)**
>
> Thanks a lot for your constructive comments!
>
> >Q1. The authors propose data destruction and adversarial outlier enhancement modules to respectively generate outliers from true samples and learn a tighter boundary. However, these two components seem not specifically aims for missing class issue in label skew one-shot fl. It would be better to further consider the relation with federated learning setting.
>
> First, our two techniques are necessary for the label-skew federated setting. As we have demonstrated in the fifth paragraph of Section 1 and Obversion 2 of Section 3.2, existing techniques such as PROSER in the centralized setting are not applicable in extreme label skews. When a client has a limited number of classes, the generated outliers are not diverse and cannot reject a great portion of samples from the unseen classes, as we see in Figure 2.
>
> Second, while our techniques are potentially applicable in the centralized setting (which is actually a benefit), they are more suitable and more important for the label-skew federated setting. We have added experiments to compare PROSER+DD+AOE and PROSER on centralized OSR settings (Appendix B.11 of the revised paper). We regard the first 2, 4, 6, and 8 classes as known during training. During testing, all 10 classes in the test set appear. We find that the improvement of DD and AOE (based on PROSER) is more significant when there are fewer classes. For PROSER, we can see a clear trend that when the number of known classes decreases, the accuracy decreases. Since PROSER generates outliers between classes, when the number of known classes is limited, the generated outliers are not diverse enough to train a good OSR model. Thus, PROSER is inapplicable to scenarios with limited known classes.
>
> Under FL label skews, some parties may have limited data for some labels due to privacy regulations and data distribution heterogeneity. Therefore, some clients may have limited known classes, e.g. only 1 class or 2 classes. Since there are multiple clients, it is unlikely to ensure that all clients have at least some number of classes. Such scenarios are different from typical centralized OSR experimental settings and pose great challenges for PROSER. Our two techniques DD and AOE can tackle these challenges under the scenario of a limited number of classes.
>
> **In conclusion, although DD and AOE can be extended to centralized OSR settings, they can bring more significant improvement to FL and are more essential for FL with label skews, compared with PROSER when the number of classes is limited, such as in FL label skew scenarios.**
>
> >Q2. The compared methods such as fedavg, fedprox, fednova are not designed for one-shot fl, it is nature that these methods perform worse on these experiments (Table1). So, it is not sure whether it is fair comparision.
>
> We compare with both one-shot FL (Close-set Voting, FedKT) and multi-round FL algorithms for comprehensive evaluation. For multi-round FL, since our setting is one-shot, we compare them with our algorithm with a single round for fair comparison.

---

> > ### Author Response · Authors · 2022-11-13
> > **Response to reviewer ZcP3 (2/2)**
> >
> > >Q3. But the proposed method is lack of novelty, seems the directly combines of existing strategies. Besides, authors do not provide in-depth rationale analysis and spend a large space of article to incorporate methodology into existing methods.
> >
> > **[Clarification of Novelty]**
> >
> > To the best of our knowledge, **we are the first to bring open-set recognition (OSR) to FL label skews, to solve the over-confidence towards seen classes.**
> > Moreover, it is non-trivial to apply the idea of open-set recognition in the federated setting.
> > **FedOV contains existing strategies but is not a naive combination. Directly applying PROSER to FL can have low accuracy in extreme label skews.** Data destruction (DD) and adversarial outlier enhancement (AOE) are proposed to solve the problem.
> > **Our proposed two techniques can significantly improve the accuracy under comprehensive label skews.** Data destruction generates diverse outliers without relying on label diversity, thus it is applicable in various label skews. Adversarial outlier enhancement further narrows down the gap between outliers and inliers, allowing the model to reject more outliers.
> >
> > Besides the two techniques, we believe that the significant improvement of introducing open-set voting into FL is valuable and insightful for the FL community. Additionally, both Reviewer hyjU and Reviewer Y1Mp think our paper is novel, technically solid and provides new perspective, which shows that our approach can bring new insights to some readers.
> >
> > **[In-depth Analysis]**
> >
> > **Visualization** We visualize the representation learned by PROSER, PROSER+DD and FedOV in Figure 2. It intuitively explains why we need them and analyzes the effect brought by our techniques DD and AOE. The following two paragraphs elaborate the intuition and reasons to design the two techniques.
> >
> > * DD operations look like data augmentation, but our idea is to randomly (1) destroy some key features and (2) keep some other features. By (1), the generated images are indeed outliers (otherwise, the model cannot differentiate inliers and outliers). By (2), the model can learn the destroyed features are indispensable for that class (otherwise, suppose you keep nothing and use random noise as outliers, the model is prone to only predict noisy images as outliers. It can predict clear images of other things as inliers). By randomness, we can destroy different key features, thus the model knows all these key features are important (otherwise, the model overfits to the feature of a specific operation). **Without any component, the trained open-set model can have a bad performance, especially for extreme label skews. We experience all these failures and summarize such criteria.**
> >
> > * DD operations are finite, while the types of outliers are infinite and cannot be simulated by a finite set of operations. AOE is proposed to simulate more outliers without expanding DD operations. The goal is to surround the inliers with a tighter boundary. By AOE, we generate outliers closer to inliers and form a better boundary to reject more outliers.
> >
> > **Effectiveness Analysis** We have added experiments in Appendix B.9 and Appendix B.11 to analyze the effectiveness of our approach. Compared with the baseline of simply applying PROSER (Appendix B.11) or FedDF (Appendix B.9), our methods bring more improvement when the label skews are more severe. Therefore, our methods are suitable for the label-skew FL setting, where some parties may have limited data for some labels due to privacy regulations and data distribution heterogeneity. Moreover, we have provided ablation studies (Section 4.3) that demonstrate the effectiveness of our proposed two techniques.

---

> > > ### Comment · Reviewer_ZcP3 · 2022-11-19
> > > **Good rebuttal**
> > >
> > > Thanks for the detailed and reasonable rebuttal. I am willing to raise my score to 6.
> > >
> > > 1. The authors conclude that DD and AOE can bring more beneficial for FL with label skews compared with centralized setting. Besides, authors confirm that existing strategies such as PROSER are applicable in extreme label skews. To some extent, it verifies the novelty of proposed method.
> > >
> > > 2. The authors provide comparison with both one-shot FL and multi-round FL. This verifies the effectiveness of proposed method. However, when comparing with FedProx and FedNova, the authors only perform single round updating. This is still unsuitable and unfair. This issue should be clarified in the final version.
> > >
> > > 3. Addressing the one-shot/few-shot federated learning is important for the community. Another paper studies the “few-shot model agnostic federated learning”. Some discussions are suggested.

---

> > > > ### Author Response · Authors · 2022-11-19
> > > > **Further response to reviewer ZcP3**
> > > >
> > > > Thank you for your response and for raising the score!
> > > >
> > > > For point 2, we will compare with multi-round FL algorithms under multiple rounds and report the experimental results in the final version if accepted.
> > > >
> > > > For point 3, we have added some discussion in the related work as follows.
> > > >
> > > > * A recent work (Huang et al., 2022) proposes few-shot model agnostic FL, which is able to train any models in a setting where each client has a very small sample size. It applies domain adaptation in the latent space with the help of a large public dataset.

---

### Author Response · Authors · 2022-11-13
**Revision summary**

We thank all reviewers for their valuable insights and suggestions to help us improve our paper. We are glad that reviewers find our FedOV is effective and achieve SOTA results verified by extensive experiments on a widely used benchmark (all reviewers), the problem studied is interesting and valuable (reviewer hyjU), our method is technically solid, novel and provides new perspective (reviwer hyjU and Y1Mp), our writing is sound, clear and easy to follow (reviwer hyjU and Y1Mp), and our code is attached for reproducibility (reviwer hyjU).

We address all the comments in the revision and the rebuttal. The major revision is summarized as follows

1. In main paper, we repeat FedDF experiments in Table 3 to answer Q5. Typos mentioned are also fixed.

2. In Appendix B.1, we add more details of Distilled FedOV to address Q6.

3. In Appendix B.9, we conduct additional experiments comparing Distilled FedOV and FedDF to further answer Q5.

4. In Appendix B.10, we discuss FL algorithms specifically for #C=1 setting and compare FedOV with FedAwS to address Q4.

5. In Appendix B.11, we evaluate our two techniques on centralized OSR experimental settings to explain why the two techniques are more suitable and more important for FL label skews for Q1.

---

### Author Response · Authors · 2022-11-17
**Looking forward to your feedbacks for our rebuttal**

Dear Reviewers,

We have addressed your concerns and revised the paper. We'd appreciate it if you could provide any feedback and raise the score if your questions are solved. Thanks for your time and effort!

---

### Decision · Program_Chairs · 2023-01-20

**Decision:**

Accept: poster

**Justification For Why Not Higher Score:**

The submission is not more competitive since its problem setting (i.e., one-shot federated learning) is very specific.

**Justification For Why Not Lower Score:**

All reviewers agreed that the novelty and significance are above the bar.

**Metareview: Summary, Strengths And Weaknesses:**

The paper studied the label skew problem for one-shot federated learning which is a problem of practical importance. It proposed an effective solution by generating outliers (with feature corruption and adversarial training techniques) and regarding the generated outliers as an additional unknown class in local training of federated learning. The authors did a very good job in the rebuttal to successfully address the concerns from the reviewers. All reviewers agreed that the novelty and significance are above the bar and thus we should accept it for publication.

**Note From Pc:**

if the above contains the word "oral" or "spotlight" please see: "oral" presentation means -> notable-top-5% and "spotlight" means -> notable-top-25%. As stated in our emails, we are disassociating presentation type from AC recommendations